# Metagenomic analysis reveals unexplored diversity of archaeal virome in the human gut

Ran Li[1,2,4], Yongming Wang[1,4], Han Hu[3], Yan Tan[3] & Yingfei Ma [1] ✉

The human gut microbiome has been extensively explored, while the archaeal viruses remain largely unknown. Here, we present a comprehensive analysis of the archaeal viruses from the human gut metagenomes and the existing virus collections using the CRISPR spacer and viral signature-based approach. This results in 1279 viral species, of which, 95.2% infect Methanobrevibacteria_A, 56.5% shared high identity (>95%) with the archaeal proviruses, 37.2% have a host range across archaeal species, and 55.7% are highly prevalent in the human population (>1%). A methanogenic archaeal virus-specific gene for pseudomurein endoisopeptidase (PeiW) frequently occurs in the viral sequences ($n = 150$). Analysis of 33 *Caudoviricetes* viruses with a complete genome often discovers the genes (*integrase*, $n = 29$; *mazE*, $n = 10$) regulating the viral lysogenic-lytic cycle, implying the dominance of temperate viruses in the archaeal virome. Together, our work uncovers the unexplored diversity of archaeal viruses, revealing the novel facet of the human gut microbiome.

The human gut microbiome is closely linked with human health[1]. In addition to the predominant bacterial component, non-bacterial members of the gut microbiota (archaea, fungi, and viruses) are known to play important roles in microbiome dynamics and human physiology, immunity, disease, etc.[2]. Archaea are also among the commensal microorganisms inhabiting other organ systems of the human body, such archaea are regularly detected in the respiratory tract, the oral cavity, and the skin[3]. Nevertheless, human-associated archaea are often overlooked and remain unconsidered, since archaea are relatively low in abundance as compared to bacteria and mostly are unculturable. As such, culture-independent methods, such as next-generation sequencing, can help capture their identity and allow a broad assessment of the human archaeome as well as the archaeal virome.

Microbial viruses exert control over the composition and metabolism of microbial communities. The dynamics of bacterial viruses in the human gut have been studied in detail so far[4,5], while few studies report the detection of the human gut archaeal viruses[6,7]. Viruses infecting archaea are notoriously diverse both in terms of their genome sequences and virion structures[8,9]. Most archaeal viruses have been thus far isolated from hyperthermophilic or halophilic hosts, with only a handful of virus species described for methanogenic and ammonia-oxidizing archaea[10]. Recent exhaustive metagenomic surveys aided the discovery of novel archaeal viruses from multiple ecosystems, including the ocean, fresh water, hot spring, and soil habitats[8]. In human feces, smacoviruses were once thought to infect eukaryotes. Recently, they were found to infect the methanogenic archaeon *Candidatus Methanomassiliicoccus intestinalis* using a CRISPR spacer-based host prediction method[11,12]. Archaeal viruses in the human gut remain highly enigmatic. Analysis of the CRISPR-Cas systems encoded by archaea revealed that 90% of all sequenced archaeal genomes hold CRISPR loci, implying a rich archaeal virome in this ecosystem[13].

The knowledge gap on archaeal viruses is fostered by the lack of their genome entries in public databases, missing conserved marker genes for viruses[7]. Only 250 archaeal viruses that infect 23 host genera

[1]Shenzhen Key Laboratory of Synthetic Genomics, Guangdong Provincial Key Laboratory of Synthetic Genomics, CAS Key Laboratory of Quantitative Engineering Biology, Shenzhen Institute of Synthetic Biology, Shenzhen Institutes of Advanced Technology, Chinese Academy of Sciences, Shenzhen 518055, China. [2]University of Chinese Academy of Sciences, Beijing 100049, China. [3]Xbiome, Scientific Research Building, Tsinghua High-Tech Park, Shenzhen, China. [4]These authors contributed equally: Ran Li, Yongming Wang. ✉e-mail: yingfei.ma@siat.ac.cn

have been described and publicly available to date[14]. These archaeal viruses are greatly diverse and the encoded proteins display very low levels of sequence homology to those in the public database[15]. Prokaryotes harbor CRISPRs to foster immunity against viruses and other invasive genetic elements, making it possible to uncover the associations between viruses and their hosts[16]. Indeed, the approach of matching the CRISPR spacers from a known organism to viruses for assigning a virus discovered by metagenomics to a host is highly reliable[17]. When viral genomic data can be linked to a specific host organism, it becomes possible to uncover novel viruses and study how they interact with their hosts within various ecosystems.

Here, we harness spacer sequences from the archaeal CRISPR-Cas systems and viral signatures to search for archaeal viruses in the human gut. First, we performed large-scale identification of archaeal genomic contigs from 2971 metagenomes derived from previously published studies (Supplementary information). Then, we obtained the spacers from the identified archaeal genomic contigs and the 1162 archaeal genomes of UHGG (Unified Human Gastrointestinal Genome)[18]. Based on the archaeal spacer collection and the signatures of protein homology present in the archaeal viruses, we established a pipeline for archaeal viral detection and obtained 1279 archaeal viral species in the human gut. This effort will contribute to a better characterization of archaeal viruses and their archaeal hosts in the human gut and provide a complementary view of the human gut microbiome.

## Result

### The human gut carries a complex, previously unexplored virome

To perform a comprehensive search for human gut archaeal viruses, first, we constructed a Human Gut Associated Archaeal Spacer Database (HGASDB) including 13,021 nonredundant CRISPR spacers recruited from the identified archaeal genomic contigs and the 1162 archaeal genomes of UHGG (Supplementary Fig. 1–3 and Supplementary Information)[18]. These spacers were derived from the contigs and genomes of different archaeal lineages, with the genus Methanobrevibacter_A contributing to the largest number of spacers (89.82%). In particular, 8962 spacers specifically were derived from Methanobrevibacter_A. smithii, 2549 spacers from Methanobrevibacter_A smithii_A, and 185 spacers from other three species (Methanobrevibacter_A woesei, Methanobrevibacter_A orals, and Methanobrevibacter_A millerae) (Supplementary Fig. 2d and Supplementary Data 1–5). A small number ($n = 1325$; 10.18%) of spacers were derived from other archaeal genera. We then identified 16,234 sequences that matched to these spacers from the 2271 assembled total community metagenomic datasets and the publicly available human gut virus collections (Fig. 1a). After we filtered out archaeal and bacterial genomic contamination and the sequences not encoding the viral signatures (i.e., hallmark genes for the known archaeal viruses) (Supplementary Fig. 4, see in Methods), these sequences were ultimately clustered (95% identity over 85% sequence) into 1279 nonredundant viral species, and the longest sequences within each species were selected as the representative in the Human Gut Archaeal Virome Database (HGAVD), for further analysis. In particular, 1080 archaeal viral representative sequences in HGAVD were detected from the assembled metagenomic datasets and 199 from other publicly available human gut virus collections (89 from IMG/VR[19], 92 from GPD[20], 14 from GVD[7], 2 from HGV[4], 1 from EVP[21], and 1 from GL-UVAB[22]). CheckV[23] analysis resulted in 12% of the sequences were classified into complete genomes (3%) and high-quality (9%) (Fig. 1b and Supplementary Data 6).

To further explore the extent to which the HGAVD viral species were homologous to the known archaeal viruses in the RefSeq database (v201) (built-in database of vConTACT2) and thereby taxonomically classify these viruses, we constructed the gene sharing networks generated by vConTACT2, where viral clusters (VCs)

approximate genus level taxonomy[24]. With the sequences from the archaeal viral genomes in the database RefSeq and the 1,279 archaeal viral species, this analysis clustered 735 HGAVD species into 61 VCs, 391 viral species into outliers (where contigs were assigned to a VC but shared fewer similar proteins than the bulk of the cluster), and 153 viral species into singletons (sequences that did not cluster with any other sequences). Only 2 VCs included one known reference viral sequence, respectively. This suggests that the majority of the VCs derived from the human gut likely represent viral genera that were novel to the viruses in RefSeq (Supplementary Data 7). Moreover, in agreement with the previous gut virome studies[20,25], the majority (68.4%) of the HGAVD viral species can't be taxonomically classified into any known viral order. Less than half of the species ($n = 404$, 31.6%) were taxonomically classified into, specifically, the *Caudoviricetes* class ($n = 389$) (tailed virus), the *Cremevirales* order ($n = 13$), and the *Haloruvirales* order ($n = 2$) (Fig. 1c). The *Cremevirales* viruses were predicted to infect *M. intestinalis* and Methanomassiliicoccus_A intestinalis, the *Haloruvirales* viruses were predicted to infect *Haloferax massiliensis*, while most of (305/389 = 78.4%) the *Caudoviricetes* species in HGAVD connected to the host of Methanobrevibacter_A smithii.

We further compared the HGAVD viruses to those of the publicly available virus collections (detailed in Method) (Fig. 1d; Supplementary Data 8, and Supplementary Fig. 5). First, we aligned the HGAVD species with the 85 nonredundant proviruses derived from 557 (50–100% completeness) of the 1162 gut archaeal genomes in UHGG[18], resulting in 56.5% ($n = 723$) of the 1279 species sharing identity >95% with those proviruses. The MGV (Metagenomic Gut Virus) catalog[25] is the newest human gut viral database and contains extensive viral genomic diversity, in particular, 102 of which was assigned to archaeal viruses. The vConTACT2 network analysis clustered the HGAVD viruses into 68 VCs, while 102 MGV archaeal virus sequences were clustered into 15 VCs, and 37 proviruses derived from the archaeal genomes in UHGG were only clustered into 9 VCs, reflecting the greater diversity of the gut archaeal virus taxa represented by HGAVD at the genus level than other virus collections. We found that a majority of the HGAVD viral species ($n = 1097$; 86%) were not clustered with any viral genomes from other collections (Fig. 1d), while a majority of 37 archaeal proviruses (78.4%) and the MGVarchaeal viral sequences (83.3%) were grouped with the HGAVD viruses, indicating that HGAVD can represent most of the archaeal viruses in other gut virus collections. Taken together, HGAVD considerably expanded the previously unknown archaeal viral diversity in the human gut.

### Archaeal viruses are highly prevalent in the human gut

We estimated the abundance of the HGAVD viral species in the human gut samples by metagenomic read recruitment (Supplementary Data 9) and accordingly performed the principal coordinate analysis (PCoA). No significant difference in the human gut archaeal viral composition was observed between male and female sex (ANOSIM, $r = 0.002$, $p = 0.306$) or according to BMI distribution (ANOSIM, $r = 0.011$, $p = 0.201$) (Supplementary Fig. 6). Nevertheless, when the analysis was stratified by country, we observed that the diversity of these archaeal viruses was distinct in the samples of different locations. In particular, the archaeal viral communities between the Tanzanian and the populations from China, America, and the UK displayed significant differences, respectively (ANOSIM, $R > 0.7$, $p < 0.001$; Fig. 2a and Supplementary Data 10).

Based on the abundance determined by the reads mapping, we further investigated the prevalence of these viruses among the human populations. The result indicated that the prevalence of 7 archaeal viral species was >10% across the human populations. These viruses belonged to 7 different VCs (Fig. 2b and Supplementary Data 7). These 7 viral species all were predicted to infect Methanobrevibacter_A smithii and had a higher prevalence in Asian, European, and American populations than in the African population. Moreover, 712 archaeal

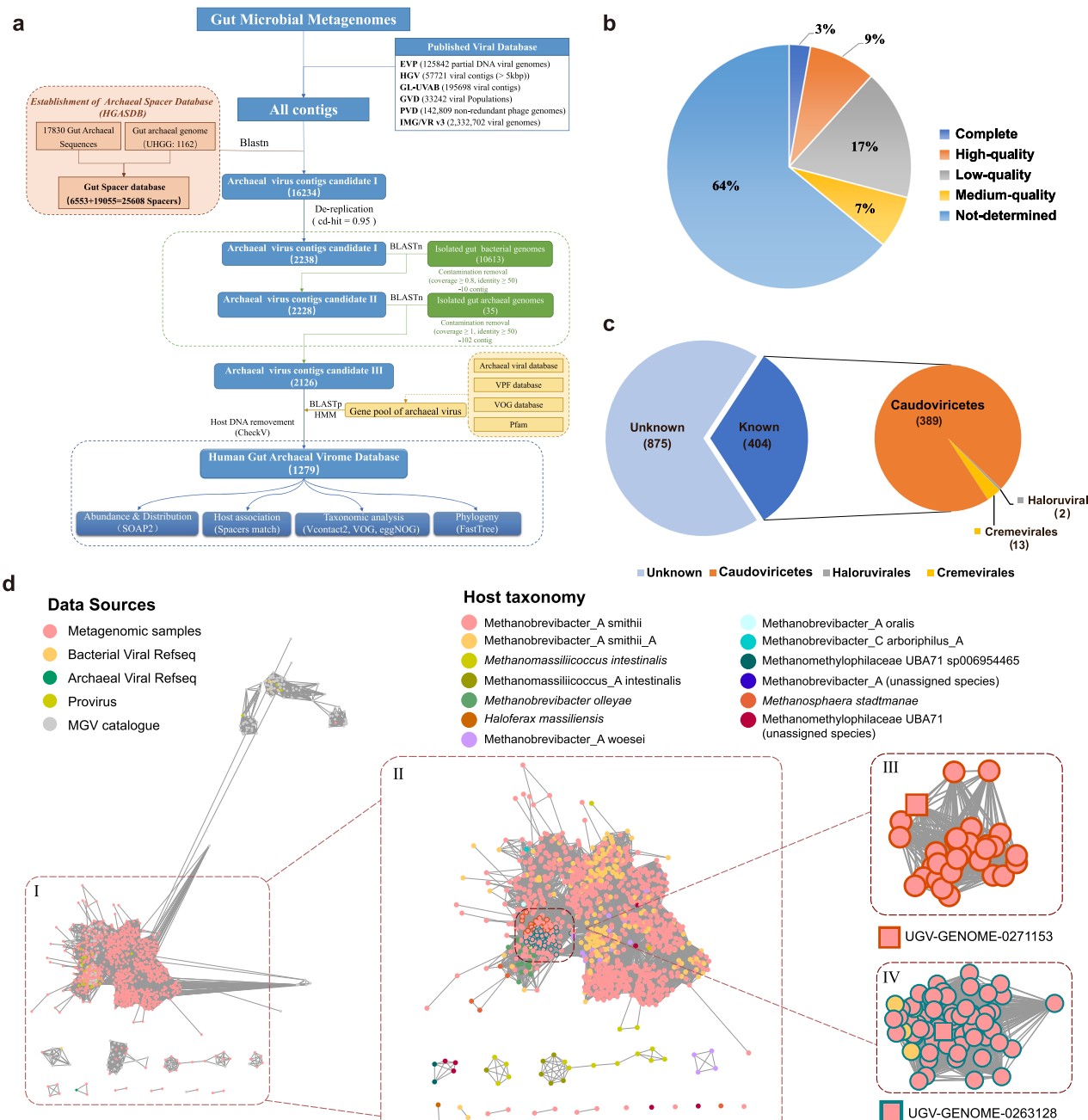

**Fig. 1 | Identification of archaeal viruses from the human gut. a** Identification workflow for archaeal viruses. See the Methods for detailed information. **b** Quality estimation of the identified viral sequences by CheckV. Evaluation of genome completeness was determined using CheckV here shown for Medium-quality ≥ 50% (MQ), High-quality ≥ 90% (HQ), and Complete = 100%. Closed genomes are annotated as "Complete". **c** Taxonomic assignment of the HGAVD viruses (order level). See the Methods for detailed information. **d** Protein clustering network of the HGAVD viruses. The network was established using vConTACT v2.0 and visualized by Cytoscape (v3.7.0) using an edge-weighted spring-embedded model. The nodes represent the viral sequences and are colored based on their sources (displayed in the legend above the network), and the width of the edges represents the number of connections between viral sequences based on shared homologous proteins. Only the viral sequences from different sources connecting with the representative sequences of the HGAVD viral species are shown (Fig. 1dI). The network containing all nodes is displayed in Supplementary Fig. 5. The viral clusters (VCs) containing the HGAVD viral species are enlarged and labeled (Fig. 1dII). Nodes are depicted in different colors representing the host taxonomy (species level) of the corresponding viral species (displayed in the legend above the network). Figure 1dIII and IV are the focused view of the network containing the two archaeal virus species with high prevalence in the human gut. Two representative contigs (IMG|UGV-GENOME-0271153 and IMG|UGV-GENOME-0263128) are shown in square shape. Source data are provided as a Source Data file.

viral species were prevalent in 1% of the human population. Remarkably, the virus IMG|UGV-GENOME-0271153, one putative medium-quality viral genome (40.51 kbp, CheckV[23]), had the highest prevalence (72.16%) among the human populations and was predicted to infect Methanobrevibacter_A smithii. This virus genome encodes 46 genes and 8 of them were predicted for the *Caudoviricetes* species functional proteins (Fig. 2c and Supplementary Data 11a). Furthermore, all the viral sequences (23–55 kbp in length) in the same VC with this virus had the host of Methanobrevibacter_A smithii (Fig. 1d) and were derived from the samples of United Kindom, Sweden, Austria, United States, China, Spain, and Madagascar, respectively, further suggesting the wide distribution of this virus among the global population. In

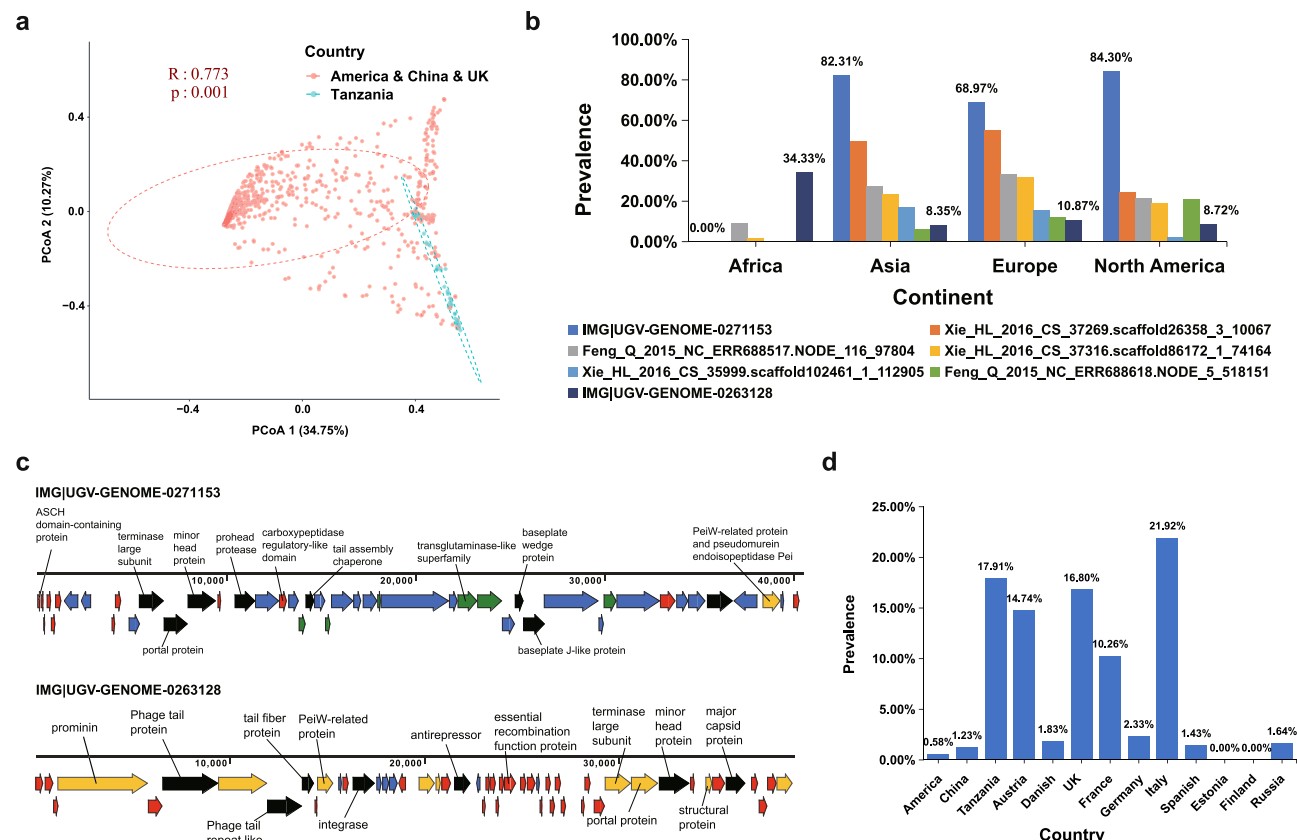

**Fig. 2 | Protein clustering network and global distribution of the HGAVD viruses in the human gut. a** PCoA of the human gut samples based on the Bray-Curtis distance matrix calculated from the abundance matrix of the HGAVD viruses. Each point is colored according to the country. R values were obtained by a two-way analysis of similarities (ANOSIM). **b** Global distribution of the HGAVD viruses with high prevalence (prevalence > 10%). **c** Genome map of IMG|UGV-GENOME-0271153 and IMG|UGV-GENOME-0263128. Genes are colored based on their best BLASTx match in the NCBI nr database. Red, the genes homologous to those of host *Methanobrevibacter_A smithii*; yellow, the genes homologous to those of other archaeal viruses; green, the genes homologous to those of other archaeal species; blue, no significant hits; black, the gene homologous to those of other viruses based on HMM analysis. Genes with a predicted function are labeled. **d** Global distribution of smacoviruses. Source data are provided as a Source Data file.

particular, another highly prevalent *Caudoviricetes* viruses (10.7%) IMG|UGV-GENOME-0263128 encoding 51 genes was detected more frequently in the African population than IMG|UGV-GENOME-0271153 (Fig. 2b). The viral sequences in the IMG|UGV-GENOME-0263128-contained VC were from 19 kbp to 56 kbp in size and were predicted to infect the hosts of Methanobrevibacter_A smithii and Methanobrevibacter_A smithii_A (Fig. 1d). These two highly prevalent viruses likely are temperate because integrase gene was detected on the genome of the virus (IMG|UGV-GENOME-0263128) or the genomes of other viruses within the same VC (IMG|UGV-GENOME-0271153) (Fig. 2c and Supplementary Data 11b).

It is worth mentioning that 13 smacovirus species were identified and were clustered into 3 VCs with lengths ranging from 2.0 to 2.5 kbp in HGAVD, reflecting the diversity of these small viruses in the human gut. Smacovirus in the order of *Cremevirales* has a small circular single-stranded DNA genome and had been identified in fecal samples (both feces and rectal swabs) of various animals[12,26]. These HGAVD smacoviruses were targeted by 7 spacers derived from the archaeal genomes in UHGG and they were predicted to infect *Methanomassiliicoccus intestinalis* or Methanomassiliicoccus_A intestinalis. Compared with the cohort of Asia and America, the prevalence of smacovirus was higher in African and European populations (Fig. 2d).

## Viruses infecting Methanobrevibacter_A smithii are a major component of the archaeal virome in the human gut

To accurately investigate the diverse virus-host interactions, we particularly screened for the CRISPR spacers present in the archaeal genomes of UHGG to target the HGAVD viral sequences. As expected, a majority (*n* = 1217; 95.2%) of the viral species connected to the genus Methanobrevibacteria_A, which is dominant in the human gut archaeaome (Fig. 3a). We then measured viral diversity by determining the number of VCs for each archaeal genera, revealing that the genus Methanobrevibacter_A harbored a significantly higher viral diversity than those of other archaeal genera (Fig. 3b), with 51 VCs assigned to this genus. Among the 51 VCs, 47 VCs were specific to Methanobrevibacter_A smithii, only 17 VCs were specific to Methanobrevibacter_A smithii_A, and 13 VCs were linked to both these two archaeal species, reflecting archaeal viruses can infect their hosts cross-species. To show this in detail, we constructed the network of host-virus by matching the HGAVD viruses with the CRISPR spacers derived from the UHGG archaeal genomes, indicating that approximately one-third of HGAVD viral species had a broad host range (Fig. 3c). Namely, 434 viral species had a host range spanning 2 archaeal species (Methanobrevibacter_A smithii and Methanobrevibacter_A smithii_A) and 12 viral species had a host range across 3 archaeal species (Methanobrevibacter_A smithii, Methanobrevibacter_A smithii_A, and Methanobrevibacter_A woesei). These analyses provide a comprehensive blueprint of archaeal virus-mediated gene flow networks in the human gut microbiome.

To further show the diversity of the tailed archaeal viruses, we searched the large subunit terminases (LST) (the marker gene for the *Caudoviricetes* viruses) from the HGAVD archaeal viral sequences and the closely related reference archaeal viruses (RefSeq database, v201) using the Pfam database, resulting in 85 LSTs derived from HGAVD

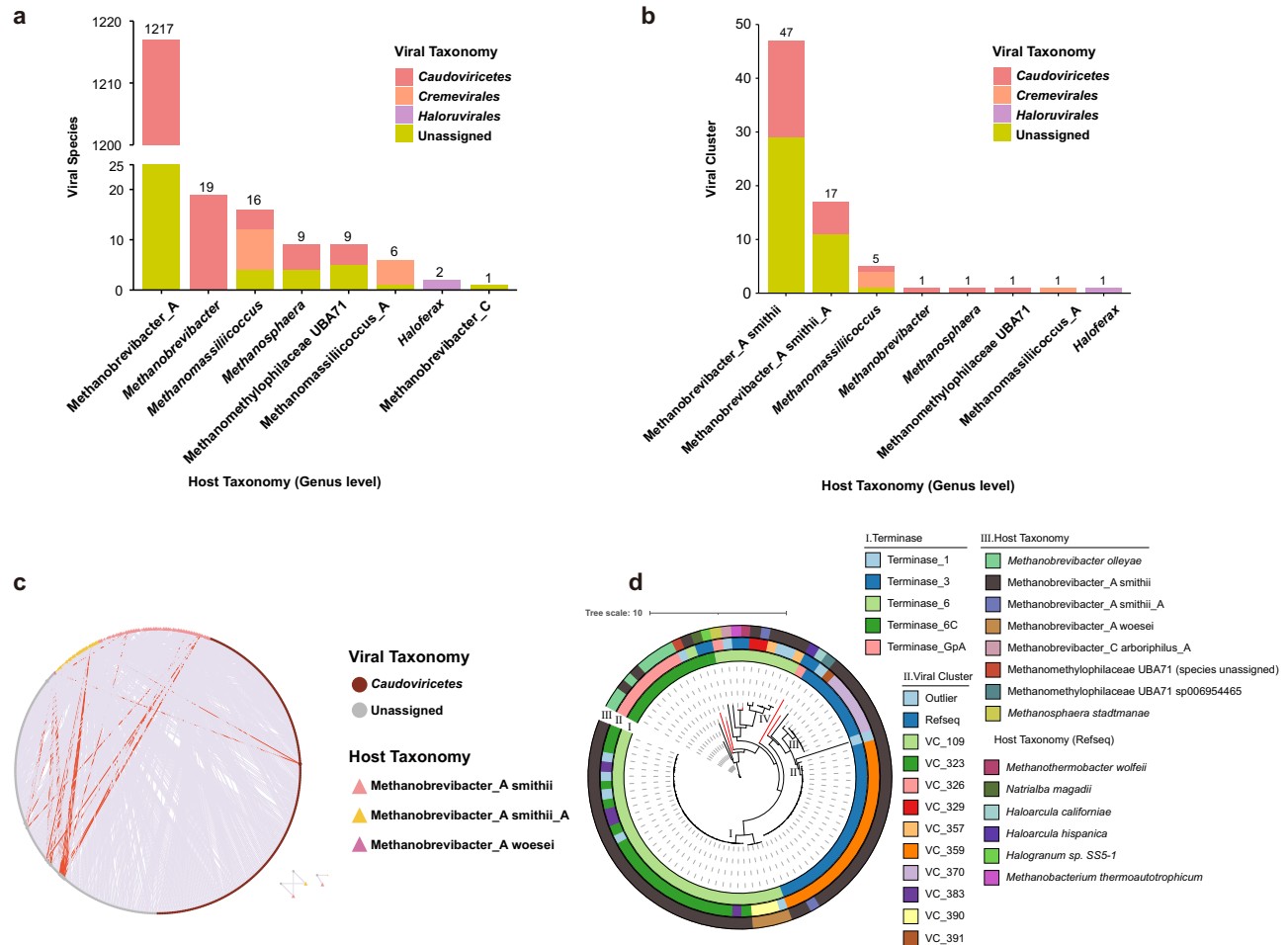

**Fig. 3 | Archaeal viral host assignment and host range determination. a** Number of the HGAVD viral species per archaeal host genus. **b** Number of viral clusters (VCs) per host genus. **c** Network showing the connections of the HGAVD viruses and their hosts. Host assignment was carried out by CRISPR spacer matching. Only the spacers recruited from the archaeal genomes of UHGG are included. The dots representing viruses are in brown, while the triangles representing the hosts of these viruses are in pink, yellow or purple. Only the connections representing viruses that link to 2 and 3 archaeal species are shown in purple and orange, respectively. **d** Phylogenetic tree of the LST protein of the HGAVD archaeal viruses. Cycle I: Pfam domains detected on the LST protein; cycle II: viral cluster (VC) that the HGAVD viruses belong to; cycle III: the viral hosts. Clades I, II, III, IV are marked in the tree. The red branches refer to LSTs of the reference viruses (GenBank No. NC_002628, NC_021328; NC_021327, NC_021322, NC_004084, NC_001902). Source data are provided as a Source Data file.

viruses belonging to at least 10 VCs and 6 homologs from 6 reference archeal viral genomes. These HGAVD LSTs were detected with 5 difference Pfam domains. The majority (68/85 = 80%) of LSTs were found encoded by the HGAVD viruses infecting the species Methanobrevibacter_A smithii, with 33 belonging to the Terminase_6 (PF03237) domain, 31 to Terminase_3 (PF04466), 3 to Terminase_6C (PF17289), and 1 to Terminase_1 (PF03354). Phylogenetic analysis of these LSTs (Fig. 3d), revealed four large gut archaeal viral clades infecting the species Methanobrevibacter_A smithii. Clade I and II without reference viruses can be defined novel clades including the largest number of HGAVD archaeal viruses. Clade III and IV had reference viruses that belong to the families *Druskaviridae* and *Leisingerviridae*, respectively, in the *Caudoviricetes* class. In conclusion, the LST phylogeny expanded the diversity of the archaeal viruses that infect Methanobrevibacter_A smithii and suggested new archaeal viral taxonomies in the human gut.

## Archaeal virus genomes encode an extensive functional repertoire

The functional potential of human gut archaea has been extensively studied[6]. HGAVD enables us to explore the functional potential of the archaeal virome in the human gut. To do this, we identified 97,208 protein-coding genes on the representative sequences of these 1279 viral species. Overall, 40% (n = 39,268) of the viral genes did not have

significant matches (cutoff: e-value < 1e-5, score > 50) in the Pfam(v32) database and were not assigned to any biological functions. Only 10.8% and 17.4% of these genes had hits in pVOG[27] and PHROG[28], respectively, indicating that remarkably little is known about the functional potential of human gut archaeal viruses (Fig. 4a and Supplementary Fig. 7).

The viruses of Methanobrevibacter_A smithii contained the most functional diversity with proteins homologous to 1,034 different kinds of tailed-virus-specific proteins in the Pfam database (only the proteins assigned biological function were taken into consideration), such as prohead protein, baseplate J, portal protein, tail fibers, and terminase large subunit, whereas other archaeal viruses lacked some of these genes (Fig. 4b and Supplementary Data 12). For example, except for the viruses infecting Methanobrevibacter_A smithii, the remainder had no proteins annotated for lysis-related functions. In particular, the genes encoding HNH endonuclease were observed on the viral genomes of both Methanobrevibacter_A smithii and Methanobrevibacter_A woesei. This protein potentially cleaves DNA into genome-length units during packaging and may operate in concert with their terminase large subunit and portal proteins[29].

The representative sequences of 36 archaeal viral species in HGAVD were measured as complete genomes by CheckV[23]. They were clustered into 7 different VCs and taxonomically classified to *Caudoviricetes* (n = 33, 6 VCs) and *Cremevirales* (n = 3, 1 VC). Analysis

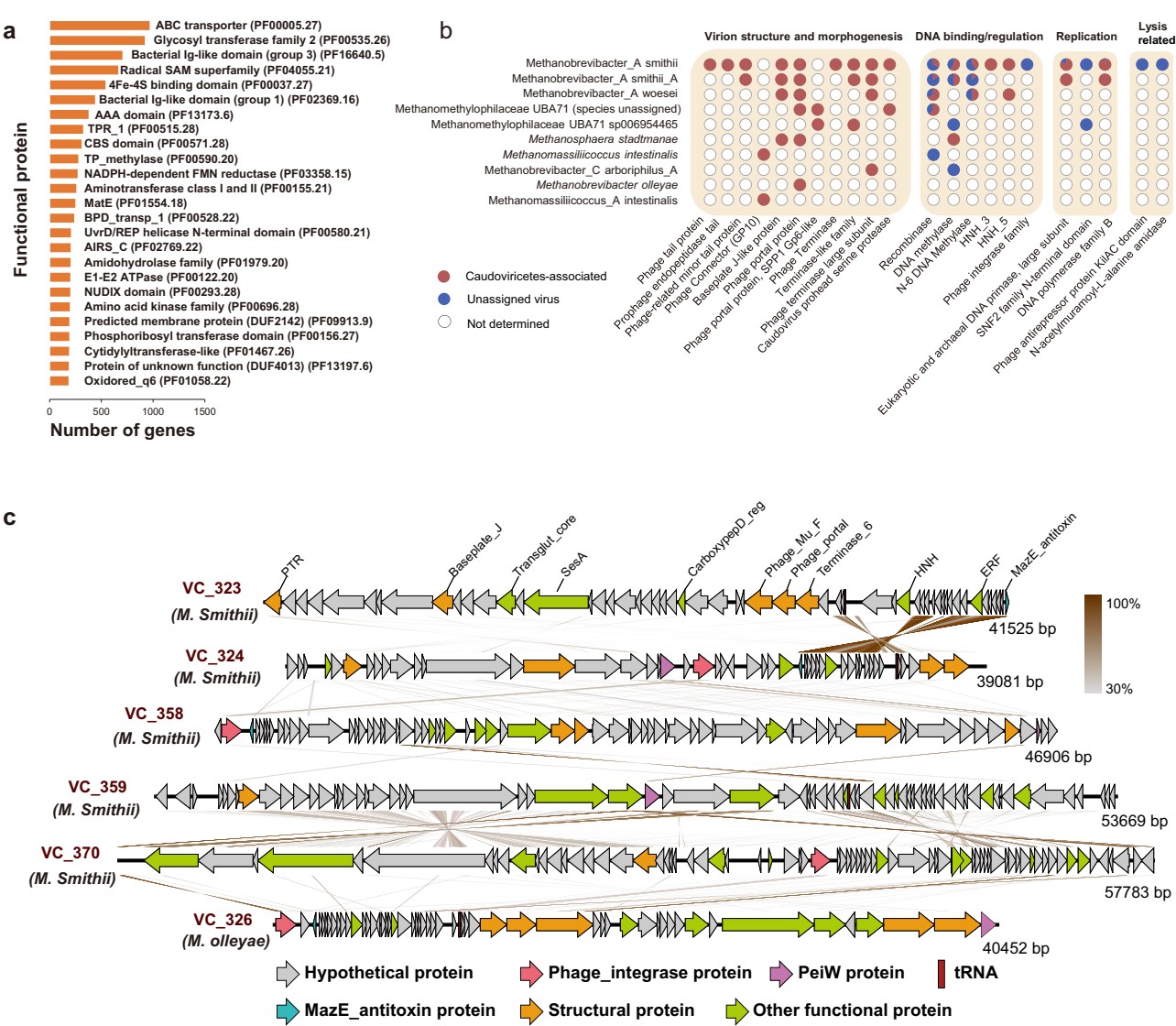

**Fig. 4 | Functional landscape of the HGAVD viruses. a** Functional annotations for the proteins encoded by the archaeal viruses. **b** Distribution of 22 genes[33] (lower x-axis) on the viruses (y-axis). Filled circles indicate annotated orthologs (red:blue ratio represent the *Caudoviricetes*:Unassigned virus ratio), while white circles indicate that the ortholog was not identified. **c** Genetic maps of the representative complete *Caudoviricetes* genomes for the 6 VCs (provir|Feng_Q_2015_NC_ERR688567.NODE_108_91642_1, provir|Feng_Q_2015_NC_ERR688611.NODE_45_142494_1, GPD|uvig_418233, IMG|UGV-GENOME-0318983, IMG|UGV-GENOME-0327529, HMP.763678604.contig63623_40452). The arrows depict the location and direction of predicted proteins on the viral genomes, and the filled colors indicate

different gene functional categories, as indicated in the legend. The annotations are based on the searches against the Pfam database, and only significant results (e-value < 1e−5) are shown. The names of the VCs are indicated in bold italic brown text. Brown shading connects genes displaying sequence similarity at the protein level, with the percent of sequence identity depicted with different shades of grey (see the scale on the right). Detailed information also can be seen in Supplementary Fig. 13. Source data are provided as a Source Data file. **c** Genetic maps of the representative complete *Caudoviricetes* genomes for the 6 VCs. We have corrected in the legend.

of these whole viral genomes in the class *Caudoviricetes* (Supplementary Data 13) resulted in an interesting finding that a gene encoding the protein homologous to pseudomurein endoisopeptidase (PeiW) frequently occurred on many viral genomes (*n* = 23). The prototype PeiW is found in the archaeal prophage psiM100 as an autolytic enzyme produced by the thermophilic methanoarchaeon *Methanothermobacter wolfeii* to cleave pseudomurein cell-wall sacculi of archaeal methanogens[30]. The phylogenetic analysis of PeiW revealed that except for the viruses of *M. wolfeii*, other archaeal viruses also were the carrier of *peiW*, such as the viruses of Methanobrevibacter_A smithii and *Methanobrevibacter olleyae* (Supplementary Fig. 8). When extending this analysis to all HGAVD viruses, 150 viruses encoded the genes of PeiW (Supplementary Fig. 9), suggesting the importance of this gene for the archaeal viruses in infecting methanogenic archaea.

In the analysis of these complete *Caudoviricetes* viral genomes, 29 of 33 encoded the genes for phage integrase protein. However, only 9 genomes were predicted as proviruses, and 20 were not flanked by host DNA by by CheckV[23]. In particular, we observed that 10 genomes infecting Methanobrevibacter_A smithii or *M. olleyae* encoded proteins belonging to the antitoxin MazE superfamily. The toxin-antitoxin system on a temperate virus acts as an addiction system, preventing the host from curing itself from the provirus[31]. Accordingly, the presence of the antitoxin MazE protein on the HGAVD archaeal viruses might highlight an arms race between the gut archaea and their viruses. Further, we performed a phylogenetic analysis based on the MazE antitoxin protein sequences detected in these viral genomes. The phylogenetic tree shows that (Supplementary Fig. 10) the viruses predicted to infect Methanobrevibacter_A smithii and *M. olleyae* were separated into different clades. We performed comparative genomic

analysis on the representative sequences selected for each VC of the complete HGAVD sequences (Fig. 4c), they were shown divergent in genomic sequence and most of the genes encoding for hypothetical proteins. Moreover, CheckV determined that only 9 genomes were predicted as proviruses and 20 were not flanked by host DNA[23], implying that most of the archaeal viruses detected in this study likely were undergoing lytic replication cycle. Overall, the analysis on these complete HGAVD viral genomes implied that temperate archaeal viruses were dominant in the human gut, similar to the human gut bacterial phages[32,33].

## Discussion

In this study, taking advantage of the metagenomic sequencing data, we conducted a comprehensive analysis of the human-associated archaeal viruses recovered from the human gut metagenomes collected worldwide, showing that the archaeal viruses were widespread in the human gut ecosystem. The results obtained in this study based on the metagenomic sequencing datasets were well-complemented with the previous study of 1167 nonredundant archaeal genomes[6]. Based on the Minimum Information about an Uncultivated Virus Genome (MIUViG) standards[34], we report the archaeal viruses related to virus origin, genome quality, functional annotation, taxonomic classification, biogeographic distribution, and host prediction. We also estimated that the average fraction of these HGAVD viruses in the human gut virome was around 0.50% (Supplementary Data 14). It has been estimated that around 1.2% of all anaerobes are human-associated archaea[6]. While the ratio of microbe:viruses is around 1:1-10 in the human gut[35], our estimation of the fraction of the HGAVD viruses in the human gut virome implied that a considerable proportion of the archaeal viruses still remain unexplored.

To date, compared to the bacterial phages, fewer archaeal viral genomes derived from the human gut were available. In the database GVD, 24 viral populations (equal to species in this study) were predicted as archaeal viruses[7]; the study related to gut archaeome reported 94 proviruses derived from the archaeal genomes[6]. These large-scale gut virus collections were conducted using several popular bioinformatic tools, such as VirSorter[36] v1.0.3, VirFinder[37] v1.1, etc. In this study the CRISPR spacer-based method, which has been widely used for linking viral and host genomes in various studies, have a better recall for the identification of previously unknown archaeal viruses[17,38,39]. In particular, analysis of previous studies indicated that more than 90% of archaea genomes harbor the CRISPR system as compared to 50% of the human gut bacterial genomes[13]. In this study, CRISPR loci were identified in 53% of the human gut archaeal genomes (including MAGs) and 80% of the isolated human gut archaeal genomes. Our stringent workflow showed a high sensitivity in identifying genome fragments for diverse gut viruses. This was evident by the detection of smacoviruses which are very small (2.5 kbp) and low abundant in the human gut microbiome. In particular, we did not detect plasmid signatures using PlasForest[40] and two sequences encoded both transposase genes and viral signatures in the HGAVD viral sequences.

While some non-viral mobile elements, such as transposons and plasmids, can also perfectly match to the spacers, these sequences were largely excluded and were not included in the HGAVD database in our workflow (Fig. 1a). In total, 847 sequences that matched to the spacers were not detected encoding genes homologous to the viral hallmark genes, 2 of which were identified as plasmid sequences, suggesting these sequences likely were derived from transposons or plasmids. Notwithstanding this, some of these excluded sequences that matched the spacers also likely represent additional families of as-yet-unidentified viruses. These novel viruses could not be identified by metagenomic approaches due to the lack of knowledge and must be determined by establishing a culture-dependent method. The isolated archaeal viruses may in turn improve the

bioinformatic methods for identifying archaeal viruses to recover more novel archaea viruses.

Taking together, in this study, we conducted a comprehensive metagenomic data mining of the archaea and the archaeal viruses in the human gut. The result revealed the diversity of the archaeal viruses and the archaea in the human gut. Considerable diversity of the unexplored archaeal viruses in the human gut and the novel viral species in HGAVD can exactly fill in the gaps in this field and serve as an expansion of the human gut archaeal viruses. Our data, together with the bacteria and bacterial phages, will provide a complementary view of the human gut virome and thus help us better understand the human gut ecosystem.

## Methods

### Collection of metagenomic sequencing data sets used for this study

Here, we collected and curated 12 human microbial metagenomic datasets consisting of 3971 human metagenomes from 1904 individuals across rural and urban populations from 13 countries (Supplementary Data 1, publicly available as of January 2021). Sequencing reads of the human gut metagenomes and the associated metadata were obtained from their respective hosting databases (e.g. SRA, iVirus, or MG-RAST). Reads were then assembled using SPAdes v3.10.0[41] with option 'meta'. The assembled contig sequences of five body sites (including the gastrointestinal tract, mouth, airways, skin, and vagina) were directly downloaded from the HMP Data Portal (https://portal.hmpdacc.org/)[42]. All the sequencing data were downloaded from online repositories or links provided in the original publications. We did not include any studies which required additional ethics committee approvals or authorizations for access.

### Detection of Archaeal genome contigs in the metagenomic sequencing datasets

The genes were predicted on the assembled contig sequences using Prodigal v2.6.3 (-p meta option)[43]. The resulting protein sequences were aligned to the Genome Taxonomy Database R95 (GTDB, R95)[44] using DIAMOND (options:−e-value 1e-3−min-score 50)[45]. According to the GTDB taxonomy system, the taxonomy of each protein was assigned based on the top hit in the database at each taxonomic rank (Phylum, order, family, genus, and species). Subsequently, Archaeal contigs were screened based on the following criteria[46]: (i) the number of encoding proteins with hit derived from archaeal genomes > the number of encoding proteins with hit derived from bacterial genomes; and (ii) the number of encoding proteins with the hits from archaeal genomes ≥ 5 (Supplementary Fig. 2a). In summary, we detected 17,830 archaeal contigs from the whole gut metagenomes and 33 archaeal contigs from other body sites (23 from the oral, 5 from the skin, and 5 from the vagina) (taxonomic information of these 33 archaeal contigs are listed in Supplementary Data 15). Meanwhile, the taxonomy of an identified archaeal contig was assigned if the number of the proteins on the contig assigned to this taxonomy was higher than others. Then all curated gut archaeal contigs sharing identity ≥95% and coverage ≥85% were dereplicated by CD-HIT v4.6[47]. Using this clustering strategy, we finally obtained 2948 nonredundant archaeal genome fragments with length >3 kbp for subsequent analysis.

### Construction of phylogenetic tree for archaeal genomes

To compare these archaeal contig sequences to the known archaeal genomes derived from the human gut, these 17,830 archaeal contigs were mapped to 1162 species-level gut archaeal genomes derived from the UHGG[18] using BLASTn (e-value ≤10-5, coverage ≥ 0.5)[48]. UHGG contains 286,997 genomes, representing 4644 species of Bacteria and Archaea from the human gut that are taxonomically annotated using GTDB-tk v.0.3.1 (GTDB R89). Taxonomy of these genomes was assigned using GTDB-Tk v0.3.3[49] based on the Genome Taxonomy

Database R202 (GTDB, http://gtdb.ecogenomic.org) taxonomy. We evaluated the quality of the genomes with CheckM[50] v1.0.11 using the 'lineage_wf' workflow. The results were further refined using maximum-likelihood phylogeny inferred from a concatenation of 122 archaeal marker genes produced by GTDB-Tk. The archaeal tree was built using RAxML v8[51] called as follows: raxmlHPCHYBRID -f a -n result -s ge input -c 25 -N 100 -p 12345 -m ROTCATLG -x 12345 and Newick tree output files were visualized with iTOL v6[52] (https://itol.embl.de/).

## Establishment of Human Gut Associated Archaeal Spacer Database (HGASDB)

The CRISPR spacer sequences were derived from two databases: (i) 17,830 gut archaeal contigs detected from the gut metagenomes, (ii) 1162 species-level archaeal genomes from the UHGG catalogue. Spacer sequences were predicted using the CRISPR Recognition Tool v1.1 (CRT)[53] with default parameters. In total, 19,055 and 6553 CRISPR spacer sequences were predicted from 1162 UHGG archaeal genomes and the 17,830 gut archaeal contigs, respectively. Redundant spacer sequences were dereplicated using CD-HIT (parameters: -c = 1, -aS = 1, -aL = 1, -g = 1), resulting in 13,021 nonredundant CRISPR spacers sequences.

## Collection reference of archaeal viral genomes

We collected a database for 202 Archaeal Viral Genomes as a reference from 3 sources:

(i) 97 reference archaeal viral genomes available in NCBI RefSeq as of December 2020.

(ii) 102 archaeal virus genomes provided in the studies of Iranzo et al.[54]. The 59 duplicated genomes compared to the genomes in (i) were removed. What's more, there were 16 genomes were labeled as "Proviruses" by Iranzo et.al. However, sequences of these proviruses have not been provided by the authors, for which reason, we used VirSorter[36] to predict the provirus from the 16 genomes. By this means, 14 proviruses have been extracted from 14 genomes. Taken together, we got 41 archaeal virus genomes from this source.

(iii) To complete the archaeal viral dataset, we included genomes of *Methanobacterium virus Drs3*[55], 43 new putative archaeal virus genomes identified from two depth profiles in the Eastern Tropical North Pacific (ETNP) oxygen minimum zone[56], 24 unknown archaeal viral populations detected by GVD[7] and 8 genomes of smacoviruses that were found to infect Archaea[11].

In total, the final archaeal virus database consisted of 202 archaeal viral genomes or fragments.

## Selection of hallmark genes for archaeal viruses

Firstly, we predicted genes from the 202 archaeal virus genomes using Prodigal v2.6.3 (default parameters) and obtained 21,985 proteins encoded by these genes. Subsequently, functional annotations were assigned to the proteins using the hmmsearch command in HMMER3 (e-value cutoff set to 1e-5)[57] against the Pfam. v. 32 database[58], a custom comprehensive viral HMM database including viral protein families (VPF) from JGI Earth's virome project[21] and the Virus Orthologous Groups (VOG) (release 202, http://vogdb.org) containing orthologous groups of numerous viruses. Then the database of archaeal viral hallmark genes was composed of the following four parts (Supplementary Fig. 4):

(1) Exclusive archaeal viral proteins based on the annotations in the Pfam database
   (i) We collected 35 genomes of archaeal isolates from UHGG catalog and each protein encoded by the genomes was annotated in the Pfam database. We selected the proteins (*n* = 1523) with the Pfam homologs only occurring on the 202 archaeal viral genomes as hallmark genes.
   (ii) If any proteins encoded by the archaeal virus genomes and the 35 isolated archaeal genomes were annotated in the Pfam database with the keywords including portal, terminase, spike, capsid, sheath, tail, coat, virion, lysin, holin, baseplate, lysozyme, head, fiber, whisker, neck, lysis, tapemeasure or structural, then these (*n* = 164) were added to the collection of hallmark genes for archaeal viruses.

(2) To include the proviruses in the archaeal genomes, we collected 11 proviruses predicted from the 35 isolated archaeal genomes in UHGG by CheckV[23] v0.6.0, and then the 249 proteins predicted from the provirus were added to the collection of the hallmark genes for archaeal viruses.

(3) The 5907 archaeal virus proteins with the best hit to the members of the VOG database were selected.

The 3368 archaeal virus proteins with the best hit to the members of the VPF database were selected.

After combining and de-replicating the proteins from these four sources, in total, 8485 proteins were selected as the hallmark genes for archaeal viruses.

## Development of archaeal viral detection workflow

To perform a comprehensive search for human gut archaeal viruses, sequences for archaeal virus detection were derived from two sources: (1) the assembled contigs of the metagenomic sequencing data we described above; 2) viral genomes identified in the published viral databases (Fig. 1a), including 125,842 partial DNA viral genomes obtained from the Earth's Virome (hereafter 'EVP')[21], 57,721 viral contigs from the Human Gut Virome database (HGV)[4], 195,698 viral contigs from Uncultured Viral Database of Archaeal and Bacteria (hereafter 'GL-UVAB')[22], 33,243 viral sequences obtained from GVD[7], 142,809 nonredundant phage genomes from GPD[20] and 2,332,702 viral genomes from IMG/VR v3[19]. To identify archaeal viral sequences from these data, we developed a viral detection workflow as follows:

(1) All the assembled metagenomic contigs were searched against HGASDB using blastn from the blast+ package v.2.2.31 (e-value < 1e-5), and 16,234 contigs that matched to the spacers were assigned as archaeal virus candidate I. These contigs were further dereplicated using the CD-HIT v4.6 with the parameters "-aS 0.85 -c 0.95". Multiple reports[7,34] have revealed that >95% ANI (Average Nucleotide Identity) was a suitable threshold for defining a set of closely related discrete 'viral group'; follow-on studies suggest that this cut-off establishes populations that are largely concordant with a biologically relevant "viral species" definition[59]. Thus, this clustering strategy resulted in 2238 viral species (represented by the longest contig within each viral species) in archaeal virus candidate I.

(2) To remove potential bacterial genome contamination, sequences of archaeal virus candidate I were queried against 16,234 isolated bacterial genomes from UHGG collection using blastn. The cut-offs defined for these searches were the minimum identity of 50%, and minimum query coverage of 80%, with a maximum e-value of $10^{-5}$. Thus 10 contigs were filtered out from candidate I and 2228 viral species remained for candidate II.

(3) To remove the contamination of archaeal genomes, sequences of archaeal virus candidate II were performed blastn against 35 isolated archaeal genomes from the UHGG collection. The cutoffs defined for these searches were the minimum identity of 50%, minimum query coverage of 100%, with maximum e-value of $10^{-5}$, Thus 102 contigs were removed from candidate II and 2126 viral species remained for candidate III.

(4) Protein sequences derived from the contigs in candidate III were compared with the protein sequences of the archaeal viral hallmark genes (identified in Selection of hallmark Genes for Archaeal Viruses) using DIAMOND. Any contigs containing best hits with a

maximum e-value of 10-5 were picked. Finally, 1279 viral species were retained for the Human Gut Archaeal Virome Database (HGAVD).

(5) For these viral species, CheckV was used to detect proviruses boundaries, remove contamination from host-derived sequences, and determine the completeness. This most recently developed tool classifies each sequence into one of five quality tiers: complete, high quality (>90% completeness), medium quality (50–90% completeness), low quality (0–50% completeness) or undetermined quality (no completeness estimated available), resulting in 12% of the sequences were classified into complete genomes (3%) and high-quality (9%) (Fig. 1b and Supplementary Data 6). In addition, we applied VirSorter (categories 1–6)[36], VirFinder (score ≥ 0.7 and $p < 0.05$)[37], VirSorter2 v2.2.3 (categories 1-6)[60] and DeepVirFinder v1.0 (score ≥ 0.9 and $p < 0.05$)[61] on the sequences in HGAVD, and in total 537 HGAVD sequences (Supplementary Data 6) were classified as viral sequences by these tools.

## Taxonomic classification of gut archaeal viruses

Two complementary approaches were used for the taxonomic classification of the 1279 archaeal viral species. First, for 1279 representative contigs of these archaeal viral species, genes were predicted using Prodigal v2.6.3 with the -p meta option. Then these predicted genes were used to cluster the 1279 archaeal viral contigs with the prokaryotic viral Refseq v201 using vConTACT v.2.0[24] with default parameters (The Refseq were supplied by the built-in database of vConTACT2). Thus, we leveraged the taxonomic information provided by the viral Refseq to taxonomically classify the contigs in these VCs. For example, if one contig in a VC is classified to the *Caudoviricetes* class, the rest contigs in this VC will also be assigned to the virus of the *Caudoviricetes* class.

Second, we used taxonomical informative profiles from the VOG database (http://vogdb.org) and eggNOG (v5.0) database[62] to find out viruses likely to be the members of the *Caudoviricetes* viral class. Specifically, we first picked out the VOGs with annotation containing the keywords (portal, terminase, spike, capsid, sheath, tail, base plate, fiber, and tape measure) and named them as Hallmark VOGs. Then the predicted proteins from the archaeal viral contigs were compared to the VOG HMM profiles and the eggNOG database using hmmsearch v3.2.1 and eggNOG-mapper v.2.0.0[63] respectively. During this process, the minimum score and maximum E-value were set to 40 and 1e-5. If the viral contig encoding genes with hits against the Hallmark VOGs or eggNOGs whose annotation contains the keywords mentioned above, then this contig will be classified into the *Caudoviricetes* viral class (Fig. 1c and Supplementary Data 6 for Order/Class-level taxonomy).

## Comparison of the viral species to other gut viral databases

A comparison between HGAVD and the viruses in publicly available databases derived from the gut microbiome was performed based on the following databases:

(i) Metagenomic Gut Virus (MGV) catalogue[25], the newest gut virus collection, contains 189,680 viral draft genomes estimated to be >50% complete and representing 54,118 candidate viral species. The protein sequences of the representative archaeal viral contigs were used as queries in a BLAST search in the MGV database with a threshold of e-value ≤ 1e-3. Only the sequences in the MGV database encoding at least one protein sequence with the hits to those of the archaeal viral contigs were retained for network analysis. (11,827/189,680 = 0.06).

(ii) Proviruses detected from 1162 gut archaeal genomes. 118 proviruses were predicted by CheckV from the 557 archaeal genome contigs in UHGG with a quality assignment of medium quality (50–90% completeness) and high quality (>90% completeness) or were complete. These proviruses were then clustered at 95% identity and 80% coverage, resulting in 85 nonredundant viral

species. We further clustered the 85 proviruses with the viruses in HGAVD. Only the 37 proviruses sharing identity ≤ 95% with the 1279 viral contigs in HGAVD were considered for further analysis.

(iii) The Prokaryotic Viral Refseq (V201) Database supplied by vConTACT2.

## Estimation of the relative abundance of viruses and hosts

First, we mapped all reads of the metagenomic sequencing data to the identified archaeal contigs and archaeal viral contigs by the software Soap2[64] v2.21, only the contigs with >30% breadth of coverage were counted. Second, the number of the reads corresponding to each of the identified archaeal genome contigs and archaeal viral contigs was normalized by the total number of the reads of each sample; the normalized value thereby represents the relative abundance of the contig in the sample.

## Estimation of the fraction of HGAVD viruses in the human gut virome

To explore the fration of the archaeal viruses in the human gut virome, we mapped raw reads collected from the 1904 samples to the 33218 non-archaeal viral sequences derived from the GVD database[7] and the HGAVD sequences by the software Soap2. The abundance of these viruses in each sample was calculated as the descriptions in the subsection of Methods "Estimation of the relative abundance of viruses and hosts."Then, we summed the abundance of archaeal viruses and bacteria viruses, respectively, and calculate the archaeal viral relative abundance in human gut virome for each sample. The average fraction of archaeal viruses in human gut virome was estimated by taking an average of the archaeal viral percentage depicted above (average: 0.50%).

## Statistical analyses

All statistical analyses were performed in R version 4.0.5. Based on Bray–Curtis dissimilarity matrices, which were calculated using the VEGAN[65] function vegdist, principal coordinate analysis (PCoA) was performed using the pcoa function in the APE package, and significant difference (p) and the degree of separation (Global R) between groups were tested by the analysis of similarities (ANOSIM) using the VEGAN function anosim. Global R ranges between 0 and 1, with Global $R = 0$ indicating no separation and Global $R = 1$ indicating complete separation. The number of permutations of anosim is 999.

## Virus-host prediction

Host-virus interactions were resolved by searching CRISPR spacer sequences in the hosts and the viral contigs. To accurately investigate the gut archaeal viruses that have a broad host range, we particularly predicted CRISPR spacers from the 1,162 archaeal genomes in the UHGG database[18] based on the following criteria: (i) CRISPR arrays were identified on the archaeal genomes longer than 10 kb using CRT[53]; (ii) To minimize spurious predictions, we dropped arrays with fewer than three spacers; (iii) CRISPR spacers were longer than 25 bp. The retained CRISPR spacers were aligned with the archaeal viral contigs using BLASTn to identify spacers present in the viral contigs, and matches satisfying the thresholds of 100% identity were selected (settings: -task blastn-short, - gapopen 10, -gapextend 2, -penalty 1, -word_size 7 -perc_identity 100).

## Phylogenetic tree analysis of genes

To construct the phylogenetic trees for large terminase subunit, PeiW and MazE-antitoxin, amino acid sequences were aligned using the MUSCLE algorithm[66] included in MEGA X[67]. The maximum-likelihood phylogenetic tree was constructed using IQ-TREE v1.6.12[68] with the automatic optimal model selection. The final consensus tree was visualized and beautified in iTOL[52].

**Reporting summary**

Further information on research design is available in the Nature Portfolio Reporting Summary linked to this article.

## Data availability

The annotated nucleotide sequences of archaeal viruses (FASTA + GFF) generated in this study, archaeal viral hallmark genes, accompanied with the metadata file describing the origin of each contig, taxonomy, including VC, host prediction information, completeness score are available in the link https://doi.org/10.6084/m9.figshare.21152404.v3. The accession codes for the sequencing data used in this study are provided in Supplementary Data 1. Source data are provided with this paper.

## Code availability

The present study did not generate code, and mentioned tools used for the data analysis were applied with default parameters unless specified otherwise.

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

## Acknowledgements

This work received support from the Strategic Priority Research Program of the Chinese Academy of Sciences (No. XDB29050500); Guangdong Provincial Key Laboratory of Synthetic Genomics (2019B030301006); Shenzhen Key Laboratory of Synthetic Genomics (ZDSYS201802061806209); Shenzhen Institute of Synthetic Biology Scientific Research Program (Grant no. JCHZ20200001)

## Author contributions

Y.M., Y.W., and R.L. designed the study. R.L. and Y.W. performed the metagenomic analysis. H.H. and Y.T. provided suggestions. Y.M., R.L., and Y.W. contributed to the scientific discussion and preparation of the manuscript.

## Competing interests

The authors declare no competing interests.
