## [Peer Review File · Nature Communications]

REVIEWER COMMENTS

Reviewer #1 (Remarks to the Author):

The study by Li et al provides an interesting new view into the archaeal virome of the human gut. Their approach to identify new archaeal viruses is rigorous and comprehensive, however it is not free from serious limitations (potentially high false-positives rate), as I discuss in more detail below. The authors have failed to provide access to their database of archaeal virus genomic contigs to enable independent verification. In addition to that, information on relative abundance of these putative viruses in viral metagenomic data from the human gut would be a very valuable addition. I believe this study is publishable after a major revision.

Line 33; What's "_A" in "Methanobrevibacteria_A"? This doesn't seem to be an officially recognised taxonomic name.

Line 34: Please expand UHGG in the abstract

Line 36: briefly define the function of *peiW* here.

Line 63: expand GTDB at first use here

Line 99: "...encoded proteins..." instead of "...encoding proteins..."

Line 109: Please expand UHGG at first use in the main text.

Line 131-133: Here and below, in accordance with International Code of Nomenclature of Prokaryotes (ICNP) genus and species names should be italicised. "Methanobrevibacter_A", "Methanomassiliicoccus_A" and "Methanocorpusculum MX-02" are not properly recognised taxonomic names. Please explain what "_A" and "MX-02" stand for.

Line 132: "Methanomethylophilaceae" appears to be a provisional name for yet unclassified clade of archaea. Should be preceded with the term "Candidatus", and not italicised. Please consult ICNP regarding the rules on this subject.

The first subsection of results "Identification of archaeal genomic contigs from the metagenomes expands the archaeal diversity in the human gut" and Fig. 1 describe a catalogue of archaeal genomic sequences identified from metagenomic datasets, which is instrumental to the main goal of the study (identification and systematisation of archaeal viruses in the gut metagenomes), but is not necessarily the main result in and of itself. I suggest moving this part to the Supplementary Information, and refer to it from line 166. The description of main study results only starts from subsection 2: "The human gut carries a complex, previously unexplored virome".

Line 170: Again what's the difference between "M. smithii" and "M. smithii_A". The second name is not a recognised taxonomic name.

Line 174: "2,271 assembled metagenomic datasets" - Do these include total community metagenomes? Virus-enriched metagenomes? Or both?

Line 175: Please indicate here whether these 16,234 sequences represent complete viral genomes or genomic fragments.

Line 177: ANI of 95% over what fraction of the sequence length? The current recommended threshold for clustering uncultured viral genomic sequences into approximately species-level groups is 95% over 85% of length (<https://doi.org/10.1038/nbt.4306>).

Line 184: As far as I know CheckV was developed primarily for assessing genomes completeness in bacterial viruses. Has it been benchmarked on archaeal viruses? Could it be that lack of the pipeline training on archaeal viruses is responsible for 67% of the genomes getting no completeness score from CheckV? In that case I question the validity of all results obtained using this tool.

Line 186: It seems that the authors used the older version of VirSorter pipeline. VirSorter2 demonstrated marked improvement of accuracy over the original version. The same applies to DeepVirFinder that superseded VirFinder. I recommend using these two newer tools to validate the results obtained using older versions.

This is an interesting observation in light of other recent independent reports, linking CRESS DNA viruses in Smacoviridae (daughter clade of CRESSvirales) to Archaea through CRISPR analysis (<https://doi.org/10.1128/2FJVI.00582-20>).

Line 206: How does this 67.9% of the HGAVD species, unclassified at order level, correspond to VirSorter/Virfinder/CheckV results described above? Could it be that the same subset was also not recognised by those virus detection pipelines? A Venn diagram would be a good illustration of the relationships between these subsets. If that was true, could it be that this 67.9% contain a significant fraction of contigs, representing non-viral mobile genetic elements (MGEs)? I question the stringency of the viral sequence selection approach chosen by the authors. From the Methods section, it seems like the presence of a CRISPR protospacer, absence of hits to bacterial and archaeal genomes, and presence of a hit (even a single hit out of many protein coding genes?) to a set of "hallmark" archaeal viral proteins, were sufficient to classify a contig as a virus. Firstly, CRISPR immunity is not unique to defence against viruses, and can be targeting other MGEs. The definition of "hallmark" proteins seems to be very relaxed as well, and even defies the purpose of being "hallmark" as it basically includes all proteins ever found on archaeal viruses (as the sub-section Selection of hallmark Genes for Archaeal Viruses in Materials suggests), whether or not they are strictly specific to the viruses, or can be shared between viruses, archaea, and their other MGEs (as the authors themselves admit in lines 442-443).

Line 206: order Caudovirales has been elevated to class Caudoviricetes in the latest edition of the ICTV taxonomy. Please consult <https://talk.ictvonline.org/taxonomy/> and correct.

Line 209: According to the current version of ICTV code, all formally accepted virus taxonomic names (kingdom to species) should be italicised. Please italicise Cromevirales and Haloruvirales here and throughout the manuscript.

Line 225: It is unclear what was done in here. Did the authors co-cluster contigs from their own HGAVD and MGVD together? In that case, "68 VCs with at least 1 HGAVD prediction" should be "68 VCs with at least 1 HGAVD-derived member".

Line 226-227: again it is unclear what are the relationships between 68 VCs from line 225 and 15 VCs and 9 VCs in here?

Fig. 3a seems to belong better with Fig. 2.

Fig. 3b. Why are America and China grouped together? Is America = USA in this context? Please clarify. UK is mentioned in the text (line 265), and other European countries are shown in Fig. 3e, but are not presented in Fig 3b. Why is that?

Line 269: It seems like the authors were only interested in differential abundance of the most prevalent viruses. Please explain why.

Line 279: "caudovirus" is neither a formally accepted taxonomic name, nor a commonly used trivial name (as the small first letter would suggest). Please change to "Caudoviricetes species", "tailed virus", or something similar.

Line 286: Correct "smacoviruse" to smacovirus.

Line 295: The sub-section is entitled "Viruses infecting *M. smithii* are a major component of the archaeal virome in the human gut". However, no information on the actual abundance (as opposed to prevalence) of these viruses in the human gut is presented. What fraction of the gut virome in each individual do they occupy? This can be measured by mapping reads from virome enriched metagenomic samples back to the contig catalogue (HGAVD) generated in this study.

Line 311: "Most of (305/388=78.6%) the caudoviral species in HGAVD connected to the host of *M. smithii*." 1) Correct caudoviral to Caudoviricetes; 2) What about Cromevirales and Haloruvirales? What archaeal hosts were they linked to? Would be nice to modify Fig. 4 a-c to provide a host taxonomy split by viral class/order.

Fig. 4d does not add much to the results described verbally in lines 313-319 and can be moved to Supplementary Information.

Line 337: I recommend to also use pVOGs and PHROG databases for this search, which focus on protein families in prokaryotic viruses.

Line 354: Is PeiW required at the stage of infection, or lysis? Does it act like endolysin in bacterial viruses? Please provide a brief explanation here.

Line 362-353: I believe this sentence is redundant. A temperate virus could be captured in the form of a provirus or a free virion in metagenomic sequencing and the following read assembly. Whether or not it was flanked by host sequences, it tells nothing about its temperate/virulent nature.

Line 365-371: In my opinion, a much simpler explanation for the presence of the complete toxin-antitoxin system on a temperate virus, is that it acts as an addiction system, preventing the host from "curing" itself from the provirus.

Line 373: "The phylogenetic tree shows that (Fig. 5e), the viruses..." Fig. 5e is not a tree.

Fig. 5b: I find it more useful to group viruses here by VC, viral class/order and the predicted host, not just the predicted host. Otherwise it is unclear what viruses are included in each line.

Fig. 5c is not cited in the text. Fig. 5c and d are redundant, as the main information is already given in lines 356-360. These can be moved to Supplementary Information.

Line 375: "...of the complete HGAVD caudoviruses (Fig. 5f)." There is no Fig. 5f.

Line 377: I wouldn't characterise the integrase gene of a temperate virus as an "accessory" gene.

Lines 377-380: "Overall, the analysis on these complete HGAVD viral genomes indicated that temperate archaeal viruses were dominant in the human gut, similar to the human gut bacterial phages, while most of the archaeal viruses detected in this study likely were in the lytic status." What do you mean by "lytic status"? Is it supposed to say that they were undergoing lytic replication cycle? Is there enough evidence to claim that?

Fig. 5e is incredibly difficult to read. These (complete?) viral genomes need to be re-oriented and circularly permuted to align to each other properly. Short protein-protein hits need to be filtered out to de-clutter the lines between genomes maps showing protein similarities. Grayscale used to highlight different percentages of similarity needs to be changed to well contrasting colours. I would appreciate addition of other representative complete or near complete genomes from the orders Cromevirales and Haloruvirales to give the reader some information about their genome structure.

Line 408: Please avoid using grandiose and self-celebrating phrases such as "...we provided unprecedented glimpses into the human archaeal virome..."

Lines 410-421 can be omitted, as they essentially repeat results section, and most importantly refer to results which are not central to this study. Focus on viruses, instead of archaeal genomic contigs.

Lines 427-428: "These tools are heavily dependent on the reference phage sequences for viral identification. This limitation caused the failure to recognize the majority of the archaeal viruses..." If that's the case, this current study is also limited in exactly the same way, as it uses databases for virus identification. This current approach is based on availability of CRISPR-spacers from currently sequenced genomes and "hallmark" genes and is also not database-independent.

Line 798: This is not a standard complete Data Availability statement. The link appears to be broken. The lab account on Github contains no information about this study. In the interest of transparency and reproducibility, the authors need to make their database of novel archaeal viruses freely available. Either by submitting the annotated contigs to GenBank, or at least by providing them with the Supplementary Information, or in one of the generalist repositories (Zenodo, Figshare etc) with a permanent DOI link. The database should contain annotated nucleotide sequences (FASTA + GFF, or GBK), accompanied with a metadata file describing the origin of each contig, taxonomy, including VC, host prediction information, completeness score etc.

Reviewer #2 (Remarks to the Author):

The archaeal component of the human gut virome is poorly characterized compared to the bacteriophages, so the study of archaeal viruses in the gut is most welcome. The work by Li et al presents such an analysis based on thousands of gut microbiome samples. I believe the identification of archaeal viruses by combining CRISPR matches and hallmark proteins was done carefully and is reliable.

What I am missing in this manuscript, is description of any truly novel viruses the authors might have discovered. It is pointed out that the majority of the detected archaeal viruses were not classifiable at the order level, which is indeed not surprising. So what is in that dark matter? I think the value and interest of the paper would increase significantly should the authors undertake and report a careful analysis of novel virus groups that are likely to be lurking there.

The manuscript presents a detailed analysis of archaeal virus abundance but as far as I can see, the most relevant information is missing, namely, a quantitative comparison with bacteriophage abundance. It is of obvious importance to know what fraction of the gut virome is comprised of archaeal viruses and whether or not that fraction is about the same as the fraction of archaea in the gut microbiome.

The manuscript is extremely detailed and seems to include many details that likely belong in the Methods section. The above, biologically more relevant and interesting analyses could be included instead.

A more technical point. It is not clear to this reviewer why the tree in Fig. 4 only includes sequences from viruses associated with a single archaeal species rather than all archaeal tailed viruses. Further, FastTree is not the best choice of the phylogenetic method. It is desirable to use a more accurate and robust method such as IQTree.

Reviewer #3 (Remarks to the Author):

This manuscript describes a straightforward approach to ID contigs belonging to archaeal viruses by using the matches to CRISPR spacers isolated from databases of gut archaeal metagenome contigs. They have been quite successful and identified 1279 archaeoviral contigs. Besides, the taxonomy of the CRISPR provides host assignment. It is a simple and useful idea and has provided a good number of bona fide archaeal viruses living in the gut microbiome. However, the analysis done on the collection of viruses does not provide relevant new information about these novel archaeal viruses. There is an excess of bioinformatic methodology but very little biological value added. The Results section is dedicated to enumeration of programs and datasets more appropriate to the methods section. The discussion is largely a repetition of the results and methodology. Some specific comments follow:

Ln127 please specify the meaning of UHGG

Ln 181 These add up to barely 200, where did the other 900 genomes come from? Please specify at least roughly

Ln 184 This is too important to give only a reference. Please specify the criteria for completeness

Ln188 442 were classified as viral, what about the remaining 800?

Ln 195 Also that the number of viral contigs being present also as proviruses is extremely high (more than 50%) which seems to indicate a very high rate of lysogeny in the gut microbiome

Ln 201 This sentence would require clarification and some of the 1279 do not fit into any of the outputs

Ln 212 Fig 2. Identification of archaeal viruses from the human gut in this work

Ln 241 Not sure what is “left panel” why not use the letters?

Ln 265 The Tanzanian datasets were from some specific cities, ethnicity, lifestyle?

Ln303 please explain the difference between M. smithii and M. smithii _A

Ln 306. Nothing surprising here. Broad host range viruses have been known for long in all domains of life

Ln 337 40% viral genes without matches are to be expected, whether or not they have a function is another issue altogether, the percentage of those should be much higher actually

Ln 345 explain “annotated for lysis”

Ln 362 how were they predicted as provirus?

Below, we provide point-by-point responses to reviewers' comments (*italicized, blue*). The corresponding revisions in the manuscript are highlighted in green.

Reviewer #1 (Remarks to the Author):

The study by Li et al provides an interesting new view into the archaeal virome of the human gut. Their approach to identify new archaeal viruses is rigorous and comprehensive, however it is not free from serious limitations (potentially high false-positives rate), as I discuss in more detail below. The authors have failed to provide access to their database of archaeal virus genomic contigs to enable independent verification. In addition to that, information on relative abundance of these putative viruses in viral metagenomic data from the human gut would be a very valuable addition. I believe this study is publishable after a major revision.

Response: We thank the reviewer for evaluating our study and appreciate these constructive and insightful comments from this reviewer.

We have clarified the questions about potentially high false-positives rate (see below the detail response to comment #17).

The access to the database of HGAVD archaeal virus genomic contigs has been repaired now (<https://doi.org/10.6084/m9.figshare.21152404.v3>).

We agree that the information on the relative abundance of putative archaeal viruses in the human gut virome is valuable, which is also raised by reviewer 2. In this revised manuscript, we conducted an experiment to estimate the relative abundance of the HGAVD archaeal viruses in the human gut virome and have added the detailed information in the revised manuscript (lines 537-545 and 341-345). Please also see the detailed information in below response to comment #27.

1) *Line 33; What's "_A" in "Methanobrevibacteria_A"? This doesn't seem to be an officially recognised taxonomic name.*

Response: In this study, the genus and species names for all the identified archaeal genomes were assigned according to the Genome Taxonomy Database (release: 04-RS89, GTDB, gtdb.ecogenomic.org) archaeal taxonomy for the domain of Archaea¹. Based on GTDB, the genus *Methanobrevibacter* has been divided into five genus-level groups: *Methanobrevibacter*, which includes the type species *Methanobrevibacter ruminantium*, and other four genera with alphabetical suffixes (*Methanobrevibacter_A* to *Methanobrevibacter_D*).

2) *Line 34: Please expand UHGG in the abstract*

Response: UHGG was expanded to Unified Human Gastrointestinal Genome² as suggested.

3) *Line 36: briefly define the function of *peiW* here.*

Response: We have revised "peiW" to "pseudomurein endoisopeptidases (PeiW)".

4) *Line 63: expand GTDB at first use here*

Response: We have revised "GTDB" to "GTDB (Genome Taxonomy Database 04-RS89)³" as suggested.

5) *Line 99: "...encoded proteins..." instead of "...encoding proteins..."*

Response: "encoding proteins" has been corrected to "encoded proteins". (L104)

6) *Line 109: Please expand UHGG at first use in the main text.*

Response: The full name of UHGG "Unified Human Gastrointestinal Genome²" has been provided.

7) *Line 131-133: Here and below, in accordance with International Code of Nomenclature of Prokaryotes (ICNP) genus and species names should be italicised. "Methanobrevibacter_A", "Methanomassiliicoccus_A" and "Methanocorpusculum MX-02" are not properly recognised taxonomic names. Please explain what "_A" and "MX-02" stand for.*

Response: We thank for this comment. All these names have been italicized in this revised manuscript.

In this study, the genus and species names for all the identified archaeal genomes were assigned according to the Genome Taxonomy Database (release: 04-RS89, GTDB, gtdb.ecogenomic.org) archaeal taxonomy for the domain of Archaea (release R04-RS89)¹. Most proposing and naming of new archaeal phyla has occurred outside the International Code of Nomenclature of Prokaryotes (ICNP) because ICNP does not consider uncultured microorganisms nor recognize the ranks of phylum and superphylum. Rinke *et al*¹ present a Genome Taxonomy Database (GTDB; gtdb.ecogenomic.org) taxonomy for the domain Archaea (release R04-RS89), comprising of 2,392 genomes from cultured and uncultured organisms. The GTDB normalizes rank assignments using relative evolutionary divergence (RED) in a concatenated protein phylogeny that takes into account differing evolutionary rates, followed by an extensive automated and manual taxonomy curation process. ICNP-declared prokaryotic type strains are recorded by NCBI taxonomists when the names have been published and are associated with sequence records submitted to the International Nucleotide Sequence Database Collaboration (INSDC)⁴. Widespread missing rank information in NCBI RefSeq89, particularly of

as-yet-uncultivated lineages, requires numerous passive (rank-filling) changes to the taxonomy. In many cases, this is not an 1:1 correspondence due to reasons including polyphyletic taxa in the NCBI taxonomy, missing assignments in the NCBI taxonomy and updated rank assignments in the GTDB taxonomy. The GTDB online browser 'taxon history' allows the comparison between GTDB and the NCBI taxa.

As such, in the case of *Methanobrevibacter_A*, based on GTDB, the genus *Methanobrevibacter* has been divided into five genus-level groups: *Methanobrevibacter*, which includes the type species *Methanobrevibacter ruminantium*, and other four genera with alphabetical suffixes (*Methanobrevibacter_A* to *Methanobrevibacter_D*). "*Methanomassiliicoccus_A*" and "*Methanocorpusculum MX-02*" stand for new uncultured archaeal genera in GTDB, respectively.

8) *Line 132: "Methanomethylophilaceae" appears to be a provisional name for yet unclassified clade of archaea. Should be preceded with the term "Candidatus", and not italicised. Please consult ICNP regarding the rules on this subject.*

Response: Thanks for the suggestion. Methanomethylophilaceae has been preceded with the term "Candidatus" and not italicised in the revised manuscript (Supplementary Information Line 20).

9) *The first subsection of results "Identification of archaeal genomic contigs from the metagenomes expands the archaeal diversity in the human gut" and Fig. 1 describe a catalogue of archaeal genomic sequences identified from metagenomic datasets, which is instrumental to the main goal of the study (identification and systematisation of archaeal viruses in the gut metagenomes), but is not necessarily the main result in and of itself. I suggest moving this part to the Supplementary Information, and refer to it from line 166. The description of main study results only starts from sub-section 2: "The human gut carries a complex, previously unexplored virome".*

Response: We appreciate this constructive suggestion. The subsection of the results "Identification of archaeal genomic contigs from the metagenomes expands the archaeal diversity in the human gut" has been moved to the Supplementary information and Fig. 1 has been moved to Supplementary Fig. 2.

10) *Line 170: Again what's the difference between "M. smithii" and "M. smithii_A". The second name is not a recognized taxonomic name.*

Response: In all this study, all the archaeal genomes were assigned names using the GTDB classification pipeline³. Pasolli et al⁵⁵ recovered 675 genomes of human gut Archaea and reconstructed its phylogeny. They found that more than half of these genomes (n = 487) belonged to the *Methanobrevibacter smithii* species-

level genome bin (ID 714), and related but diverged species-level genome bin including 94 genomes was identified (ID 713, 5.6% nucleotide divergence from the *M. smithii* isolate genome). Chibani et al⁶ report the analysis of public archaeal genomes from the human gut, and propose splitting the most abundant archaeal species, *Methanobrevibacter smithii*, into two species-level clades (tentatively named ‘*smithii*’ and ‘*smithii_A*’ according to the GTDB classification³, based on an ANI of only 93.95% between the two representative genomes of *M. smithii* and *M. smithii_A*, as well as information derived from the protein catalogue. *M. smithii_A* and *M. smithii* cannot be distinguished on 16S ribosomal RNA gene sequences, which is most probably the reason for missing this clade separation previously. However, analysis of the *mcrA* gene revealed a consistent difference between these two clades, with an average of 2.15% difference in amino acid sequence.

According to the GTDB classification taxonomy, both *M. smithii* and *M.smithii_A* belong to the genus *Methanobrevibacter_A*. Therefore, we use *Methanobrevibacter_A smithii* and *Methanobrevibacter_A smithii_A* for these two species in this revised manuscript.

11) *Line 174: "2,271 assembled metagenomic datasets" - Do these include total community metagenomes? Virus-enriched metagenomes? Or both?*

Response: The 2,271 assembled metagenomic datasets only included total community metagenomes, because in our experiment, we first needed to identify the archaeal genome for recruitment of the spacers derived from archaeal genomes. We describe the datasets in the section of Materials and Methods (“Collection of metagenomic sequencing data sets used for this study”).

12) *Line 175: Please indicate here whether these 16,234 sequences represent complete viral genomes or genomic fragments.*

Response: In the identification workflow for archaeal viruses (Fig. 1a, in this revised manuscript), these 16,234 sequences were only recognized by the HGASDB(Human Gut Archaeal Spacer database) spacers from the 2,271 assembled metagenomic datasets and the publicly available human gut virus collections (EVP⁷, HGV⁸, GL-UVAB⁹, GVD¹⁰, PVD and IMG/VR¹¹). **Therefore, these sequences include viral sequences, other mobile genetic elements (MGEs), archaeal and bacterial genomic fragments.** Then, we filtered out archaeal and bacterial genomic contaminations by querying against the genomes of 10,613 isolates from UHGG (Unified Human Gastrointestinal Genome)² collection using blastn and the sequences not encoding the viral signatures, which are likely MGEs. The remaining sequences were ultimately clustered into 1,279 nonredundant viral sequences.

We have rephrased the sentence to clarify this: “*We then identified 16,234 sequences that matched to these spacers from the 2,271 assembled total community metagenomic datasets and the publicly available human gut virus collections (Fig. 1a)*” (lines 111-113)

13) *Line 177: ANI of 95% over what fraction of the sequence length? The current recommended threshold for clustering uncultured viral genomic sequences into approximately species-level groups is 95% over 85% of length (<https://doi.org/10.1038/nbt.4306>).*

Response: In our study, the parameters were set as ANI of 95% over 85% of the sequence length to cluster the sequences based on the suggestion of the Uncultivated Virus Genome (MIUViG) standards¹².

14) *Line 184: As far as I know CheckV was developed primarily for assessing genomes completeness in bacterial viruses. Has it been benchmarked on archaeal viruses? Could it be that lack of the pipeline training on archaeal viruses is responsible for 67% of the genomes getting no completeness score from CheckV? In that case I question the validity of all results obtained using this tool.*

Response: The reviewer is correct that the tool is developed for assessing viral genome completeness. In the paper of CheckV¹³, this tool has been benchmarked on archaeal proviruses and has been used to detect bacterial and archaeal host contaminations on provirus sequences. Moreover, Chibani *et al*⁶ also used this tool to estimate completeness and assess the quality of archaeal provirus sequences, resulting in 45 high-quality and 130 medium-quality archaeal viral genomes.

As a most recently developed software tool, CheckV is an automated pipeline for identifying closed viral genomes and estimating the completeness of viral genome fragments. CheckV estimates viral genome completeness by comparing sequences with a large database of complete viral genomes available in the public database¹³ (NCBI GenBank, IMG/M¹⁴, Global Ocean Virome 2.0 dataset¹⁵ et al.). However, due to the limitation of the database of complete viral genomes and the majority of unexplored viruses in the human gut, this tool does not work well on the unexplored archaeal viruses in the human gut due to the lack of the reference viruses in the database. **As mentioned in the paper of CheckV¹³, authors highlight that it will be important to incorporate new viral genomes as these become available to continually expand the diversity of the reference database.**

This tool has been applied to the databases IMG/VR¹¹ and GOV2.0¹⁵, resulting in 89.3% and 92.5% low quality or undetermined sequences. In this study, it is not surprising that **these sequences (67%) were not determined the viral score by this tool. These sequences likely are novel archaeal viruses to those references in the database of CheckV**, as we can see that 56.5% (n=723) showed high similarity (ANI>95%)

to the proviruses predicted from the archaeal genomes (lines 126-130). **On the other side, due to the highly fragmental nature of the sequences assembled from metagenomic sequences in this study, these viral sequences that are fragmental can not be recognized either by this tool.**

Nevertheless, the result of this tool provided useful estimation of our predicted data, as we can see that the complete viral genomes assigned by CheckV are all typical tailed viruses similar to the tailed bacterial phages. This enables us to characterize these archaeal viruses with complete genomes identified by this tools (lines 300-332), further providing insights into the human gut archaeal virome.

15) *Line 186: It seems that the authors used the older version of VirSorter pipeline. VirSorter2 demonstrated marked improvement of accuracy over the original version. The same applies to DeepVirFinder that superceded VirFinder. I recommend using these two newer tools to validate the results obtained using older versions.*

Response: We thank the reviewer for this suggestion.

In this revised manuscript, we applied VirSorter⁵ (categories 1, 2, 4, 5, 6), VirFinder⁶ (score ≥ 0.9 and $p < 0.05$), VirSorter2¹⁶ and DeepVirFinder¹⁷ (score ≥ 0.9 and $p < 0.05$) to the sequences in HGAVD, and the number of the HGAVD sequences (Supplementary Table 6) that were classified as viral sequences by these tools increased to 537 in total (line 125). This result has been updated in our revised manuscript. We also show a Venn diagram to compare the identification efficacy of these software tools (Response Fig. 1).

Response Fig. 1 Venn diagram showing the number of the viral sequences classified by four tools

16) *This is an interesting observation in light of other recent independent reports, linking CRESS DNA viruses in Smacoviridae (daughter clade of Cremevirales) to Archaea through CRISPR analysis (<https://doi.org/10.1128%2FJVI.00582-20>).*

Response: Thank you for your valuable reference on Smacoviridae and the paper has been cited.

17) *Line 206: How does this 67.9% of the HGAVD species, unclassified at order level, correspond to VirSorter/Virfinder/CheckV results described above? Could it be that the same subset was also not recognized by those virus detection pipelines? A Venn diagram would be a good illustration of the relationships between these subsets. If that was true, could it be that this 67.9% contain a significant fraction of contigs, representing non-viral mobile genetic elements (MGEs)? I question the stringency of the viral sequence selection approach chosen by the authors. From the Methods section, it seems like the presence of a CRISPR protospacer, absence of hits to bacterial and archaeal genomes, and presence of a hit (even a single hit out of many protein coding genes?) to a set of "hallmark" archaeal viral proteins, were sufficient to classify a contig as a virus. Firstly, CRISPR immunity is not unique to defence against viruses, and can be targeting other MGEs. The definition of "hallmark" proteins seems to be very relaxed as well, and even defies the purpose of being "hallmark" as it basically includes all proteins ever found on archaeal viruses (as the sub-section Selection of hallmark Genes for Archaeal Viruses in Materials suggests), whether or not they are strictly specific to the viruses, or can be shared between viruses, archaea, and their other MGEs (as the authors themselves admit in lines 442-443).*

Response: We appreciate these very insightful comments. These comments reveal multiple aspects that we should further clarify. Please see below information.

The majority of the HGAVD viral species were not taxonomically classified into known viral orders using the available reference viral sequences in public databases. This case is in agreement with those of other gut virome studies using the similar viral taxonomical classification method⁹⁻¹¹. For example, in GPD database, around 80% of GPD VCs are not assigned a viral family. It is not surprising that the majority of phage sequences uncovered by metagenomics could not be classified into any known viral taxonomy laid out by the International Committee on Taxonomy of Viruses (ICTV)¹⁸.

We described the detailed information of taxonomic assignment for HGAVD viral species in the section of Materials and Methods (“Taxonomic classification of gut archaeal viruses”) that was developed in this study according to the method described in the study of GPD with slight modifications. Briefly, we assigned the taxonomy of the HGAVD viruses using two complementary approaches. **One way is to cluster the HGAVD viral sequences with the prokaryotic viral Refseq using vConTACT.** The taxonomic information of the viral RefSeq was assigned to the HGAVD viruses in the same VCs. **The other way is to find if the HGAVD viral sequences encoding genes with hits against the hallmark VOGs (<http://vogdb.org>) or eggNOGs¹⁹ whose functions are critical for the viruses in the Caudoviricetes viral class.** Apparently, these

reference-based approaches are limited to assign taxonomy for novel viruses due to the great divergence of viral genomes in nature. Although a majority of the HGAVD viral sequences were not taxonomically classified into known viral order, we compared the HGAVD viral sequences with the proviruses derived from the gut archaeal genomes in UHGG², resulting in 56.5% of the HGAVD viruses sharing >95% identity with the proviruses⁶. This suggests that many HGAVD sequences that were not taxonomically assigned into known viral order are archaeal viruses.

We applied VirSorter, VirFinder, VirSorter2, CheckV and DeepVirFinder to the classified and unclassified subsets of the HGAVD viruses. The Venn diagrams showing the number of shared and unique viral sequences between these subsets identified by these tools were provided as suggested (Response Fig. 2). In this figure, we can see that many HGAVD viral sequences that were not assigned taxonomically were identified as viruses by these tools. This result shows the various capability in identifying viral sequences by these tools. We also applied these tools to the complete archaeal viral genomes (n=216) downloaded from the NCBI's Nucleotide database (GenBank), as described previously¹⁸, resulting in 18 viral genomes that were not identified by these tools and around 50% (n=107) were identified as low-quality or undetermined by CheckV¹³. This shows the limitation of these tools in identifying novel archaeal viruses.

Response Fig. 2 Venn diagrams showing the number of shared and unique (a) unknown and (b) known taxonomy viral sequences classified by five tools.

Our method that we developed to identify archaeal viruses from the metagenomic sequencing dataset is fairly rigorous. Our method was developed based on the method in the study of GPD⁶ with modifications. **We did not include all proteins ever found on the archaeal viruses in the selection of hallmark genes for archaeal viruses.** The hallmark genes only include **I**)the proteins with homologs specific to the 202 archaeal

viral genomes that we recruited from the NCBI RefSeq database; **II**) for the proteins that were found on both the 35 archaeal genomes and the 202 archaeal virus genomes, only the proteins that were annotated as the virus-specific functions (portal, terminase, spike, capsid, sheath, tail, coat, virion, lysin, holin, baseplate, lysozyme, head, fiber, whisker, neck, lysis, tapemeasure or structura were included; **III**) the proteins encoded by the 11 proviruses predicted from the 35 isolated archaeal genome were included; **IV**) the proteins encoded by the 202 archaeal viral genome with the best hit to the members of the VOG (virus orthologous groups) database were included; **V**) and the proteins with the best hit to the members of the VPF (viral protein family) database were included. Moreover, to remove potential bacterial and archaeal genome contamination, the sequences that matched to the HGASDB spacers were aligned to the 16,234 gut bacterial genomes and 35 isolated archaeal genomes to filter out the contaminated sequences that have high similarity with the bacterial and archaeal genomic sequences. **In addition, we did not detect plasmid signatures using PlasForest²⁰ in these sequences. Two sequences encoded both transposase genes and viral signatures.** Although we clustered the sequences and selected the longest sequences as representative sequences for further analysis, a number of sequences are partial fragments of the viruses due to the fragmental nature of the metagenomic assembled sequences, also likely leading to the failure of detection by these tools.

It is worth mentioning that we have excluded the sequences (n=847) that were perfectly matched to the spacers but were not detected encoding genes homologous to the viral hallmark genes in the collection of candidate II (Fig.2b). These sequences likely were derived from transposons or plasmids and were excluded from the HGAVD collection, since we detected two plasmid contigs in these excluded sequences using PlasForest²⁰, and two sequences encoded transposase or conjugative transposon genes. **These sequences without viral signature hits were not included in the HGAVD database.**

Therefore, for the HGAVD viruses that were not assigned to viral orders in this study, the reasons for this can be attributed to 1) the majority of the archaeal viruses are novel and have no similarity to the references of archaeal viruses deposited in public databases; 2) due to the fragmental nature of the metagenomic assembled sequences, these identified sequences are only fragments of some whole viral genome sequences and do not encode viral marker genes for taxonomic assignment using bioinformatic approaches. In addition, **our analysis demonstrated the advantage of the method using the CRISPR-spacers and viral signatures in identifying novel viruses, particularly, the human gut archaeal viruses that have few reference viral genomes in public databases.** These novel viruses could not be taxonomically assigned due to the lack of reference sequences and must be determined by establishing a culture-dependent method or deep sequencing method. The isolated archaeal viruses may in turn improve the bioinformatic methods for identifying archaeal viruses to recover more novel archaeal viruses.

18) *Line 206: order Caudovirales has been elevated to class Caudoviricetes in the latest edition of the ICTV taxonomy. Please consult <https://talk.ictvonline.org/taxonomy/> and correct.*

Response: Thanks for the suggestion. We have revised "order Caudovirales" to "class Caudoviricetes" throughout the paper.

19) *Line 209: According to the current version of ICTV code, all formally accepted virus taxonomic names (kingdom to species) should be italicised. Please italicise Cremevirales and Haloruvirales here and throughout the manuscript.*

Response: We thank the reviewer for this comment. We have italicized *Cremevirales* and *Haloruvirales* throughout the manuscript.

20) *Line 225: It is unclear what was done in here. Did the authors co-cluster contigs from their own HGAVD and MGVD together? In that case, "68 VCs with at least 1 HGAVD prediction" should be "68 VCs with at least 1 HGAVD-derived member".*

Response: Yes, we co-clustered the viral sequences collected in HGAVD, the MGVD (Metagenomic Gut Virus) catalog²¹, and the proviruses derived from the archaeal genomes of UHGG together using the vConTACT2 network to compare the diversity of archaeal viruses collection in these databases. To clarify this, we have rephrased the description in the revised manuscript “*The vConTACT2 network analysis clustered the HGAVD viruses into 68 VCs, while 102 MGVD archaeal virus sequences were clustered into 15 VCs, and 37 proviruses derived from the archaeal genomes in UHGG were only clustered into 9 VCs, reflecting the greater diversity of the gut archaeal virus taxa represented by HGAVD at the genus level than other virus collections. We found that a majority of the HGAVD viral species (n=1,097; 86%) were not clustered with any viral genomes from other collections (Fig. 1d), while a majority of 37 archaeal proviruses (78.4%) and the MGVD archaeal viral sequences (83.3%) were grouped with the HGAVD viruses, indicating that HGAVD can represent most of the archaeal viruses in other gut virus collections (lines 151-158)*”.

21) *Line 226-227: again it is unclear what are the relationships between 68 VCs from line 225 and 15 VCs and 9 VCs in here?*

Response: We performed this analysis to reflect the greater diversity of the gut archaeal virus taxa represented by HGAVD at the genus level (VC) than those of archaeal MGVD viruses and proviruses identified from

archaeal genomes. To clarify this, we have updated the sentences in the revised manuscript (lines 151-158, also see the detailed information in the response to comment #21).

22) *Fig. 3a seems to belong better with Fig. 2.*

Response: Fig. 2 has been updated. We have moved Fig. 3a to Fig. 2d as you suggested. We thank for this suggestion.

23) *Fig. 3b. Why are America and China grouped together? Is America = USA in this context? Please clarify. UK is mentioned in the text (line 265), and other European countries are shown in Fig. 3e, but are not presented in Fig 3b. Why is that?*

Response: Yes, America refers to USA. The community composition of HGAVD viruses in the human gut from USA, UK, and China was not significantly different (Supplementary Table 10) while we found that the archaeal viral communities between the Tanzanian and the populations from China, America, and the UK displayed significant differences, respectively (ANOSIM, $R > 0.7$, $p < 0.001$; Fig. 2a and Supplementary Table 10). The Tanzanian dataset was sampled from Hadza people in Africa which are a group of people far away from modern civilization²². This implies that the living habitat can impact the human gut microbiome. Fig. 3e (Fig.2 d, in the revised manuscript) displays the prevalence of smacovirus across 13 countries, whereas Fig. 3b (Fig. 2a) only depicts the countries that show significant community composition differences based on the relative abundance of the HGAVD viruses in the human gut. Supplementary Table 10 shows the ANOSIM (analysis of similarity) statistics of community composition of archaeal viruses between different countries.

24) *Line 269: It seems like the authors were only interested in differential abundance of the most prevalent viruses. Please explain why.*

Response: We thank the reviewer for this comment. Indeed, we did analyze the abundance and prevalence of all HGAVD viruses in the human gut, 712 of which were prevalent in 1% of the human population (Supplementary Table 9). Here, we only show the most prevalent archaeal viruses (>10%) that are more representative than others in Figure 2b, and the detailed information of all viruses are shown in Supplementary Table 6.

25) *Line 279: "caudovirus" is neither a formally accepted taxonomic name, nor a commonly used trivial name (as the small first letter would suggest). Please change to "Caudoviricetes species", "tailed virus", or something similar.*

Response: “caudovirus” has been corrected to “Caudoviricetes species” or “tailed virus” in this revised manuscript. (L306). We thank the reviewer for this suggestion.

26) *Line 286: Correct "smacoviruse" to smacovirus.*

Response: “smacoviruse” has been corrected to “smacovirus” throughout the manuscript. Thanks.

27) *Line 295: The sub-section is entitled "Viruses infecting M. smithii are a major component of the archaeal virome in the human gut". However, no information on the actual abundance (as opposed to prevalence) of these viruses in the human gut is presented. What fraction of the gut virome in each individual do they occupy? This can be measured by mapping reads from virome enriched metagenomic samples back to the contig catalogue (HGAVD) generated in this study.*

Response: We greatly appreciate this very insightful and constructive suggestion.

Indeed, we have estimated the relative abundance of these viral sequences in the samples (Supplementary Table 9) (Methods “Estimation of the relative abundance of viruses and hosts”).

In this revised manuscript, we conducted an experiment as suggested to estimate the relative abundance of 1279 gut archaeal viruses in the gut virome. We mapped raw reads collected from the human gut metagenomic samples (n=1904) to the 33218 non-archaeal viral sequences derived from the GVD database¹⁰ and the HGAVD sequences by the software Soap2²³. The abundance of these viruses in each sample was calculated as the descriptions in the subsection of Materials and Methods “Estimation of the relative abundance of viruses and hosts”. Then, we summed the abundance of archaeal viruses and bacterial viruses, respectively, and calculated the archaeal viral relative abundance in human gut virome for each sample. The average relative abundance of archaeal viruses in the human gut virome was estimated as 0.50%. Specifically, the archaeal viruses contributed to 0.21%, 0.66%, 0.76% and 0.29% of the human gut virome from Asia, Europe, Africa and North America populations, respectively. We did not detect any HGAVD sequences in 142 samples. The relative abundance of archaeal viruses in human gut virome from 158 samples was more than 1%. Detailed information on “The relative abundance of archaeal viruses in human gut virome” has been added in the Materials and Methods subsection and Supplementary Table 14 as suggested. In the human gut, it is estimated that around 1.22% of all anaerobes are human-associated archaea^{6, 24}. Since these is an estimation that the ratio of microbe:viruse is around 1:1-10²⁵, in this study, the average estimated relative

abundance of the HGAVD viruses in human gut virome is around 0.50%, implying that there may be many archaeal viruses to be unexplored.

We have updated this information to the revised manuscript (lines 341-345).

28) *Line 311: "Most of (305/388=78.6%) the caudoviral species in HGAVD connected to the host of M. smithii." 1) Correct cudoviral to Caudoviricetes; 2) What about Cremevirales and Haloruvirales? What archaeal hosts were they linked to? Would be nice to modify Fig. 4 a-c to provide a host taxonomy split by viral class/order.*

Response: “cudoviral” has been corrected to “Caudoviricetes”.

A more detailed description is now added in the Results section (lines 144-147): “Less than half of the species (n=404) were taxonomically classified into, specifically, the *Caudoviricetes* class (n=388), the *Cremevirales* order (n = 13), and the *Haloruvirales* order (n =2) (Fig. 1c). The *Cremevirales* viruses were predicted to infect *M. intestinalis* and *Methanomassiliicoccus_A intestinalis*, and the *Haloruvirales* viruses were predicted to infect *Haloferax massiliensis*, while most of (n=305) the *Caudoviricetes* species in HGAVD connected to the host of *Methanobrevibacter_A smithii*”. We have updated this information in lines 141-147.

All Figures were substantially re-worked as suggested in Fig.3 a, b and c in this revised manuscript.

29) *Fig. 4d does not add much to the results described verbally in lines 313-319 and can be moved to Supplementary Information.*

Response: We thank the reviewer for this suggestion.

To reveal more information about the human gut archaeal viruses regarding the HGAVD viral sequences, in this revised manuscript, we re-analyzed all the large subunit terminase sequences (n=85) recruited from the HGAVD viral sequences and constructed the phylogenetic tree. Therefore, we updated Fig. 4d as Fig. 3d to show more biological information about these human gut archaeal viruses. This information has been updated in the revised manuscript (lines 250-263). Please see below:

“To further show the diversity of the tailed archaeal viruses, we searched the large subunit terminases (LST) (the marker gene for the Caudoviricetes viruses) from the HGAVD archaeal viral sequences and the closely related reference archaeal viruses (RefSeq database, v201) using the Pfam database, resulting in 85 LSTs derived from HGAVD viruses belonging to at least 10 VCs and 6 homologs from 6 reference archeal viral genomes. These HGAVD LSTs were detected with 5 difference Pfam domains. The majority (68/85=80%) of LSTs were found encoded by the HGAVD viruses infecting the species Methanobrevibacter_A smithii, with 33 belonging to the Terminase_6 (PF03237) domain, 31 to Terminase_3 (PF04466), 3 to Terminase_6C

(PF17289), and 1 to *Terminase_1* (PF03354). Phylogenetic analysis of these LSTs (Fig. 3d), revealed four large gut archaeal viral clades infecting the species *Methanobrevibacter_A smithii*. Clad I and II without reference viruses can be defined novel clades including the largest number of HGAVD archaeal viruses. Clad III and IV had reference viruses that belong to the families Druskaviridae and Leisingerviridae, respectively, in the Caudoviricetes class. In conclusion, the LST phylogeny expanded the diversity of the archaeal viruses that infect *Methanobrevibacter_A smithii* and suggested new archaeal viral taxonomies in the human gut.”

Response Fig. 3 (Fig. 3d) Phylogenetic tree of the LST protein of the HGAVD archaeal viruses. Out cycle: Pfam domains detected on the LST protein; Middle cycle: viral cluster (VC) that the HGAVD viruses belong to; Inner cycle: the viral hosts. The red branches refer to LST of the reference viruses (GenBank No. NC_002628, NC_021328; NC_021327, NC_021322, NC_004084, NC_001902)

30) *Line 337: I recommend to also use pVOGs and PHROG databases for this search, which focus on protein families in prokaryotic viruses.*

Response: We appreciate this suggestion. We have compared protein-coding genes on the representative sequences for these 1,279 viral species with the pVOGs¹⁸ and PHROG²⁶ databases, Overall, only 10.8% and 17.4% of the viral genes have significant matches (cutoff: e-value < 1e-5) in the pVOGs and PHROG database, respectively, which also indicates that remarkably little is known about the functional potential of human gut archaeal viruses.

We have updated this information in the revised manuscript in lines 276-277.

31) *Line 354: Is PeiW required at the stage of infection, or lysis? Does it act like endolysin in bacterial viruses? Please provide a brief explanation here.*

Response: Yes. PeiW identified from archaeal proviruses acts as autolysins for the pseudomurein cell wall. Pseudomurein, the major cell wall component of methanogenic archaea²⁷. We added this information “*The prototype PeiW is found in the archaeal prophage psiM100 as an autolytic enzyme to cleave pseudomurein cell-wall sacculi of archaeal methanogens*²⁸ in lines 293-295”.

32) *Line 362-353: I believe this sentence is redundant. A temperate virus could be captured in the form of a provirus or a free virion in metagenomic sequencing and the following read assembly. Whether or not it was flanked by host sequences, it tells nothing about its temperate/virulent nature.*

Response: We feel sorry for the confusion to this reviewer.

In this study, we hope to determine if the temperate viruses were undergoing lytic replication cycle. Because the HGAVD viruses were recruited from the metagenomic sequencing dataset, the metagenomic sequencing method can capture the temperate viruses that are derived from their host genomes or the temperate viruses that are undergoing lytic replication cycle. We did not determine the viral genomes (n=20) flanked by host DNA, implying that these temperate viruses likely were undergoing lytic replication cycle.

33) *Line 365-371: In my opinion, a much simpler explanation for the presence of the complete toxin-antitoxin system on a temperate virus, is that it acts as an addiction system, preventing the host from "curing" itself from the provirus.*

Response: Thanks for this suggestion. We have updated the information in the revised manuscript: “*The toxin-antitoxin system on a temperate virus acts as an addiction system, preventing the host from curing itself from the provirus. Accordingly, the presence of the antitoxin MazE protein on the HGAVD archaeal viruses might highlight an arms race between the gut archaea and their viruses*”.

34) *Line 373: "The phylogenetic tree shows that (Fig. 5e), the viruses..." Fig. 5e is not a tree.*

Response: We apologize for this mistake. This used to be Fig. 5d. In this revised manuscript, Fig.5d has been moved to Supplementary Fig.11

35) *Fig. 5b: I find it more useful to group viruses here by VC, viral class/order and the predicted host, not just the predicted host. Otherwise, it is unclear what viruses are included in each line.*

Response: We appreciate this insightful suggestion.

In this revised Figure, besides the predicted hosts, we also show the viral class/orders that the function proteins were derived from. Since the HGAVD viruses can be classified into numerous VCs, we show the detailed VC information for these viruses in Supplementary Table 12 (Distribution of genes on the VCs). We conducted this annotation in the Pfam database, while the viruses of the orders *Cremevirales* and *Haloruvirales* were not annotated in the Pfam database.

Response Fig. 3 (Fig. 4b) Distribution of 22 genes (lower x-axis) on the viruses (y-axis). Filled circles indicate annotated orthologs (red: the Caudoviricetes; blue: Unassigned viruses; while: not determined).

36) *Fig. 5c is not cited in the text. Fig. 5c and d are redundant, as the main information is already given in lines 356-360. These can be moved to Supplementary Information.*

Response: We apologize for this mistake. Fig. 5c has been cited in the revised manuscript now. Fig. 5c and Fig. 5d have been moved to Supplementary Figure as suggested. (Supplementary Fig. 9 and Fig. 11).

37) *Line 375: "...of the complete HGAVD caudoviruses (Fig. 5f)." There is no Fig. 5f.*

Response: We apologize for this mistake. This used to be Fig. 5e. We have corrected the figure legend in the revised manuscript (Fig. 4c).

38) *Line 377: I wouldn't characterise the integrase gene of a temperate virus as an "accessory" gene.*

Response: The “integrase” has been removed from here as you suggested.

39) *Lines 377-380: "Overall, the analysis on these complete HGAVD viral genomes indicated that temperate archaeal viruses were dominant in the human gut, similar to the human gut bacterial phages, while most of the archaeal viruses detected in this study likely were in the lytic status." What do you mean by "lytic status"? Is it supposed to say that they were undergoing lytic replication cycle? Is there enough evidence to claim that?*

Response: Yes. Here, "lytic status" refers to that the viruses were undergoing lytic replication cycle.

In this study, we observed that 29 of 33 HGAVD viruses with complete genomes predicted by CheckV encoded the phage integrase gene or MazE gene and 10 of 33 shared high identity (ANI>95%) with the proviruses derived from the gut archaeal genomes⁶. This implies that the majority of these viruses are temperate viruses. Because the HGAVD viruses were recruited from the metagenomic sequencing dataset, the metagenomic sequencing method can capture the temperate viruses that are derived from their host genomes or the temperate viruses that are undergoing lytic replication cycle. However, CheckV predicted only 9 genomes as proviruses by checking if the genomes were flanked by host DNA¹³ and 20 were not flanked by host DNA (lines 301-302). This likely implies that they were undergoing lytic replication cycle.

Currently, we don't have adequate evidence to support this claim based on the bioinformatic method. In this revised manuscript, we rephrase the sentence to "*Moreover, CheckV determined that only 9 genomes were predicted as proviruses and 20 were not flanked by host DNA¹³, implying that most of the archaeal viruses detected in this study likely were undergoing lytic replication cycle. Overall, the analysis on these complete HGAVD viral genomes implied that temperate archaeal viruses were dominant in the human gut, similar to the human gut bacterial phages²⁹.*" (lines 312-316)

40) *Fig. 5e is incredibly difficult to read. These (complete?) viral genomes need to be re-oriented and circularly permuted to align to each other properly. Short protein-protein hits need to be filtered out to de-clutter the lines between genomes maps showing protein similarities. Grayscale used to highlight different percentages of similarity needs to be changed to well contrasting colors. I would appreciate addition of other representative complete or near complete genomes from the orders *Cremevirales* and *Haloruvirales* to give the reader some information about their genome structure.*

Response: We appreciate this suggestion. We have modified the Fig. 5e (Fig. 4c in the revised manuscript). Short protein-protein hits have been filtered out by setting the cutoff of the protein similarity (> 30%). The Grayscale has also been changed to more contrasting colors. What's more, the genetic maps of the representative genomes from the orders *Cremevirales* and *Haloruvirales* are shown in the Supplementary

Figures (Supplementary Fig. 12 and Fig. 13) and the genetic maps for all complete genome HGAVD viruses are shown in Supplementary Fig. 14)

Response Fig. 4 (Fig. 4c) Genetic maps of the representative complete Caudoviricetes genomes for the 6 VCs.

Response Fig. 5 (Supplementary Fig. 12) Genetic map of the representative genome from the order Cremevirales in HGAVD. The arrows depict the location and direction of predicted genes on the viral genome, and the filled colors indicate different gene functional categories, as depicted in the legend. The annotations were based on searches against the nr database, and only significant results (e -value $< 1e-5$) are shown.

Response Fig. 6 (Supplementary Fig. 13) Schematic genomic alignment of the two representative genome of the order *Haloruvirales* in HGAVD and the *Halogeometricum pleomorphic virus 1* HGPV-1 (GenBank accession JN882267) as linear representation. HGPV-1 genome is used as a reference. The arrows depict the location and direction of predicted genes on the viral genome, and the filled colors

indicate different gene functional categories. The annotations were based on searches against the nr GenBank database, and only significant results (e-value < 1e-5) are shown.

41) *Line 408: Please avoid using grandiose and self-celebrating phrases such as "...we provided unprecedented glimpses into the human archaeal virome..."*

Response: Thank you for the suggestion. We have removed this sentence from here in this revised manuscript.

42) *Lines 410-421 can be omitted, as they essentially repeat results section, and most importantly refer to results which are not central to this study. Focus on viruses, instead of archaeal genomic contigs.*

Response: We thank the reviewer for this suggestion. The statements in lines 410-421 have been deleted from here.

43) *Lines 427-428: "These tools are heavily dependent on the reference phage sequences for viral identification. This limitation caused the failure to recognize the majority of the archaeal viruses..." If that's the case, this current study is also limited in exactly the same way, as it uses databases for virus identification. This current approach is based on availability of CRISPR-spacers from currently sequenced genomes and "hallmark" genes and is also not database-independent.*

Response: We thank this reviewer for this comment. We have rephrased the statement to highlight the advantage of CRISPR spacer-based method in identifying the novel archaeal viruses from the human gut: *"These large-scale gut virus collections were conducted using several popular bioinformatic tools, such as VirSorter⁵, VirFinder⁶, etc. In this study the CRISPR spacer-based method, which has been widely used for linking viral and host genomes in various studies, have better recall for the identification of previously unknown archaeal viruses."* (lines 348-351).

44) *Line 798: This is not a standard complete Data Availability statement. The link appears to be broken. The lab account on Github contains no information about this study. In the interest of transparency and reproducibility, the authors need to make their database of novel archaeal viruses freely available. Either by submitting the annotated contigs to GenBank, or at least by providing them with the Supplementary Information, or in on of the generalist repositories (Zenodo, Figshare etc) with a permanent DOI link. The database should contain annotated nucleotide sequences (FASTA + GFF, or GBK), accompanied*

with a metadata file describing the origin of each contig, taxonomy, including VC, host prediction information, completeness score etc.

Response: We thank the reviewer for catching this error. The annotated nucleotide sequences of novel archaeal viruses (FASTA + GFF) are available in figshare with the DOI link <https://doi.org/10.6084/m9.figshare.21152404.v3>). A metadata file describing the origin of each archaeal viral contig (taxonomy, VC, host prediction information, completeness score etc.) were provided as Supplementary Table 6 as well.

We thank the reviewer again for providing these very insightful comments again.

Reviewer #2 (Remarks to the Author):

The archaeal component of the human gut virome is poorly characterized compared to the bacteriophages, so the study of archaeal viruses in the gut is most welcome. The work by Li et al presents such an analysis based on thousands of gut microbiome samples. I believe the identification of archaeal viruses by combining CRISPR matches and hallmark proteins was done carefully and is reliable.

Response: We thank the reviewer for recognizing the values of our study and for providing insightful comments.

1) What I am missing in this manuscript, is description of any truly novel viruses the authors might have discovered. It is pointed out that the majority of the detected archaeal viruses were not classifiable at the order level, which is indeed not surprising. So what is in that dark matter? I think the value and interest of the paper would increase significantly should the authors undertake and reported a careful analysis of novel virus groups that are likely to be lurking there.

Response: We thank the reviewer for evaluating our manuscript. These comments reveal multiple aspects that we should further clarify.

The majority of the HGAVD viral species were not taxonomically classified into known viral order. This case is in agreement with those of other gut virome studies^{10, 21}. For example, in GPD (gut phage database) database, around 80% of GPD VCs are not assigned a viral family. It is not surprising that the majority of phage sequences uncovered by metagenomics could not be classified into any known viral taxonomy laid out by the International Committee on Taxonomy of Viruses (ICTV)³⁰. As such, in this study, the reasons for this can be attributed to 1) the majority of the archaeal viruses are novel and have no similarity to the references of archaeal viruses deposited in public databases; 2) due to the fragmental nature of the metagenomic assembled sequences, these identified sequences are only fragments of some whole viral genome sequences and do not encode viral marker genes for taxonomic assignment using bioinformatic approaches.

In this study, although we were not able to assign the majority of the HGAVD viruses at the order/class level, we conducted the analysis of the archaeal viral sequences related to virus origin, genome quality, functional annotation, taxonomic classification, biogeographic distribution, and host prediction based on the Minimum Information about an Uncultivated Virus Genome (MIUViG) standards¹². Leveraging this comprehensive archaeal viral sequence collection, we provided some glimpses into the human archaeal virome, leading to a better understanding of the human gut ecosystem.

We summarize the major findings of this study:

1) The number of the archaeal taxa was expanded to 4 phyla, 8 families, 22 genera, and 56 species in the human gut;

2) We identified 1,279 archaeal viral species, most (n=1,217; 95.2%) of which were specific to infect the genus *Methanobrevibacter*_A.

3) Among the viruses that can be assigned to viral taxonomy, the viral species that belong to the class *Caudoviricetes* (n = 389) (virus characterized by having tails and icosahedral capsids) are dominant, followed by the *Cremevirales* order (n = 13), and the *Haloruvirales* order (n =2); the *Cremevirales* viruses were predicted to infect *M. intestinalis* and *Methanomassiliicoccus*_A *intestinalis*, the *Haloruvirales* viruses were predicted to infect *Haloferax massiliensis*, while most of (305/389=78.4%) the *Caudoviricetes* species in HGAVD connected to the host of *Methanobrevibacter*_A *smithii*.

4)The prevalence of these archaeal viral species was significantly distinct across different countries and the prevalence of a virus infecting *Methanobrevibacter*_A *smithii* reached 72.16% in the global human population;

5)These archaeal virus genomes encode an extensive function repertoire. In particular, in the analysis of 36 complete viral genomes, we observed that an archaeal virus-specific gene *peiW* and the genes (for integrase and *MazE*) regulating the viral lysogenic-lytic cycle frequently occurred on these genomes, implying the dominance of temperate viruses in the human gut archaeal virome.

6) We conducted additional experiment to estimate that the fraction of the HGAVD viruses in human gut virome is around 0.50%, implying that there may be many archaeal viruses that remain unexplored

Our analysis demonstrated the advantage of the method using CRISPR-spacers and viral signatures in identifying novel viruses, particularly the archaeal viruses in the human gut. To further characterize these novel archaeal viruses, more cutting-edge methods are needed, including deep sequencing and long-read sequencing methods to obtain the extremely low abundant viruses in the human gut, as well as culture-dependent methods to isolate the viruses for archaeal bacterial isolates, which will be shown in future studies.

2)The manuscript presents a detailed analysis of archaeal virus abundance but as far as I can see, the most relevant information is missing, namely, a quantitative comparison with bacteriophage abundance. It is of obvious importance to know what fraction of the gut virome is comprised of archaeal viruses and whether or not that fraction is about the same as the fraction of archaea in the gut microbiome.

Response: We greatly appreciate this very insightful and constructive suggestion. We agree that the information of the relative abundance of the archaeal viruses in the gut virome is valuable.

Indeed, we have estimated the relative abundance of these viral sequences in the samples (Supplementary Table 9) (Methods “Estimation of the relative abundance of viruses and hosts”).

In this revised manuscript, we conducted an experiment as suggested to estimate the relative abundance of 1279 gut archaeal viruses in the gut virome. We mapped raw reads collected from the human gut metagenomic samples (n=1904) to the 33218 non-archaeal viral sequences derived from the GVD database¹⁰ and the HGAVD sequences by the software Soap2²³. The abundance of these viruses in each sample was calculated as the descriptions in the subsection of Materials and Methods “Estimation of the relative abundance of viruses and hosts”. Then, we summed the abundance of archaeal viruses and bacterial viruses, respectively, and calculated the archaeal viral relative abundance in human gut virome for each sample. The average relative abundance of archaeal viruses in the human gut virome was estimated as 0.50%. Specifically, the archaeal viruses contributed to 0.21%, 0.66%, 0.76% and 0.29% of the human gut virome from Asia, Europe, Africa and North America populations, respectively. We did not detect any HGAVD sequences in 142 samples. The relative abundance of archaeal viruses in human gut virome from 158 samples was more than 1%. Detailed information on “The relative abundance of archaeal viruses in human gut virome” has been added in the Materials and Methods subsection and Supplementary Table 14 as suggested. In the human gut, it is estimated that around 1.22% of all anaerobes are human-associated archaea^{6, 24}. Since these is an estimation that the ratio of microbe:viruse is around 1:1-10²⁵, in this study, the average estimated relative abundance of the HGAVD viruses in human gut virome is around 0.50%, implying that there may be many archaeal viruses yet to be explored.

We have updated this information to the revised manuscript (lines 341-345).

2) *The manuscript is extremely detailed an seems to include many details that likely belong in the Methods section. The above, biologically more relevant and interesting analyses could be included instead.*

Response: We greatly appreciate this very constructive suggestion. We have revised the manuscript as suggested to highlight the biological information in this study. Please see the revised manuscript with modifications marked in green.

3) *A more technical point. It is not clear to this reviewer why the tree in Fig. 4 only includes sequences from viruses associated with a single archaeal species rather than all archaeal tailed viruses. Further, FastTree is not the best choice of the phylogenetic method. It is desirable to use a more accurate and robust method such as IQTree.*

Response: We thank the reviewer for the comments and the suggestions.

To reveal more information about the human gut archaeal viruses regarding the HGAVD viral sequences, in this revised manuscript, we re-analyzed all the large subunit terminase sequences (n=85) recruited from all tailed HGAVD viral sequences and constructed the phylogenetic tree as suggested. Therefore, we updated Fig. 4d as Fig. 3d to show more biological information about these human gut archaeal viruses. This information has been updated in the revised manuscript (lines 250-263). Please see below:

“To further show the diversity of the tailed archaeal viruses, we searched the large subunit terminases (LST) (the marker gene for the Caudoviricetes viruses) from the HGAVD archaeal viral sequences and the closely related reference archaeal viruses (RefSeq database, v201) using the Pfam database, resulting in 85 LSTs derived from HGAVD viruses belonging to at least 10 VCs and 6 homologs from 6 reference archaeal viral genomes. These HGAVD LSTs were detected with 5 difference Pfam domains. The majority (68/85=80%) of LSTs were found encoded by the HGAVD viruses infecting the species *Methanobrevibacter_A smithii*, with 33 belonging to the Terminase_6 (PF03237) domain, 31 to Terminase_3 (PF04466), 3 to Terminase_6C (PF17289), and 1 to Terminase_1 (PF03354). Phylogenetic analysis of these LSTs (Fig. 3d), revealed four large gut archaeal viral clades infecting the species *Methanobrevibacter_A smithii*. Clad I and II without reference viruses can be defined novel clades including the largest number of HGAVD archaeal viruses. Clad III and IV had reference viruses that belong to the families Druskaviridae and Leisingerviridae, respectively, in the Caudoviricetes class. In conclusion, the LST phylogeny expanded the diversity of the archaeal viruses that infect *Methanobrevibacter_A smithii* and suggested new archaeal viral taxonomies in the human gut.”

Response Fig. 7 (Fig. 3d) Phylogenetic tree of the LST protein of the HGAVD archaeal viruses. Cycle I: Pfam

domains detected on the LST protein; cycle II: viral cluster (VC) that the HGAVD viruses belong to; cycle III: the viral hosts. Clades I, II, III, IV are marked in the tree. The red branches refer to LSTs of the closely related reference viruses (GenBank No. NC_002628, NC_021328; NC_021327, NC_021322, NC_004084, NC_001902).

All other trees have been updated using the method IQtree.

Response Fig. 8 (Supplementary Fig. 9) Maximum likelihood phylogenetic tree of PeiW proteins encoded by the complete genomes of the HGAVD viral species. The tree was constructed using the automatic optimal model selection. The star shows the prototype protein of PeiW (UniProtKB Q7LYX0).

Response Fig. 9 (Supplementary Fig. 10) Maximum likelihood phylogenetic tree of PeiW proteins encoded by all HGAVD viruses. The tree was constructed using the automatic optimal model selection. The star shows the prototype protein of PeiW (UniProtKB Q7LYX0).

Response Fig. 10 (Supplementary Fig. 11) Maximum likelihood phylogenetic tree of MazE-antitoxin proteins encoded by the complete genomes of the HGAVD viral species.

We thank the reviewer again for providing these very insightful comments again.

Reviewer #3 (Remarks to the Author):

This manuscript describes a straightforward approach to ID contigs belonging to archaeal viruses by using the matches to CRISPR spacers isolated from databases of gut archaeal metagenome contigs. They have been quite successful and identified 1279 archaeal viral contigs. Besides, the taxonomy of the CRISPR provides host assignment. It is a simple and useful idea and has provided a good number of bona fide archaeal viruses living in the gut microbiome. However, the analysis done on the collection of viruses does not provide relevant new information about these novel archaeal viruses. There is an excess of bioinformatic methodology but very little biological value added. The Results section is dedicated to enumeration of programs and datasets more appropriate to the methods section. The discussion is largely a repetition of the results and methodology.

Response: Many thanks for these constructive suggestions and comments. These comments reveal multiple aspects that we should further clarify.

The majority of the HGAVD viral species were not taxonomically classified into known viral order. This case is in agreement with those of other gut virome studies^{10, 21}. For example, in GPD (gut phage database) database, around 80% of GPD VCs are not assigned a viral family. It is not surprising that the majority of phage sequences uncovered by metagenomics could not be classified into any known viral taxonomy laid out by the International Committee on Taxonomy of Viruses (ICTV)³⁰. As such, in this study, the reasons for this can be attributed to 1) the majority of the archaeal viruses are novel and have no similarity to the references of archaeal viruses deposited in public databases; 2) due to the fragmental nature of the metagenomic assembled sequences, these identified sequences are only fragments of some whole viral genome sequences and do not encode viral marker genes for taxonomic assignment using bioinformatic approaches.

In this study, although we were not able to assign the majority of the HGAVD viruses at the order/class level, we conducted the analysis of the archaeal viral sequences related to virus origin, genome quality, functional annotation, taxonomic classification, biogeographic distribution, and host prediction based on the Minimum Information about an Uncultivated Virus Genome (MIUViG) standards¹². Leveraging this comprehensive archaeal viral sequence collection, we provided some glimpses into the human archaeal virome, leading to a better understanding of the human gut ecosystem.

We summarize the main findings of this study:

1) The number of the archaeal taxa was expanded to 4 phyla, 8 families, 22 genera, and 56 species in the human gut;

2) We identified 1,279 archaeal viral species, most (n=1,217; 95.2%) of which were specific to infect the genus *Methanobrevibacteria*_A.

3) Among the viruses that can be assigned to viral taxonomy, the viral species that belong to the class *Caudoviricetes* (n = 389) (virus characterized by having tails and icosahedral capsids) are dominant, followed by the *Cretevirales* order (n = 13), and the *Haloruvirales* order (n = 2); the *Cretevirales* viruses were predicted to infect *M. intestinalis* and *Methanomassiliicoccus_A intestinalis*, the *Haloruvirales* viruses were predicted to infect *Haloferax massiliensis*, while most of (305/388=78.6%) the *Caudoviricetes* species in HGAVD connected to the host of *Methanobrevibacter_A smithii*.

4) The prevalence of these archaeal viral species was significantly distinct across different countries and the prevalence of a virus infecting *Methanobrevibacter smithii* reached 72.16% in the global human population;

5) These archaeal virus genomes encode an extensive function repertoire. In particular, in the analysis of 36 complete viral genomes, we observed that an archaeal virus-specific gene *peiW* and the genes (for integrase and *MazE*) regulating the viral lysogenic-lytic cycle frequently occurred on these genomes, implying the dominance of temperate viruses in the human gut archaeal virome.

6) We conducted additional experiment to estimate that the fraction of the HGAVD viruses in human gut virome is around 0.50%, implying that there may be many archaeal viruses that remain unexplored

Our analysis demonstrated the advantage of the method using CRISPR-spacers and viral signatures in identifying novel viruses, particularly the archaeal viruses in the human gut. To further characterize these novel archaeal viruses and overcome, more cutting-edge methods are needed, including deep sequencing and long-read sequencing methods to obtain the extremely low abundant viruses in the human gut, as well as culture-dependent methods to isolate the viruses for archaeal bacterial isolates, which will be shown in future studies.

We have revised the results and discussion sections of manuscript as suggested and conducted additional experiments to highlight the biological information of the archaeal viruses. All changes is marked in green in this revised manuscript.

Some specific comments follow:

1) *Ln127 please specify the meaning of UHGG.*

Response: UHGG refers to Unified Human Gastrointestinal Genome². It represents a database that contains 286,997 bacterial and archaeal genomes derived from the human gut. The section “Identification of archaeal genomic contigs from metagenomes expands the archaeal diversity in the human gut.” has been moved to Supplementary Information as suggested by reviewer 1.

2) *Ln 181 These add up to barely 200, where did the other 900 genomes come from? Please specify at least roughly.*

Response: We thank for this suggestion. Specifically, the archaeal viral representative sequences identified from published viral databases add up to 19 (89 from IMG/VR¹¹, 92 from GPD, 14 from GVD¹⁰, 2 from HGV⁸, 1 from EVP⁷, and 1 from GL-UVAB⁹), and the other 1,080 genomes derived from the assembled metagenomic datasets we collected.

To perform a comprehensive search for human gut archaeal viruses, sequences for archaeal virus detection were derived from two sources: 1) the assembled contigs of the metagenomic sequencing data we collected; 2) viral genomes identified in the published viral databases, including the viral genomes obtained from the Earth's Virome (EVP)⁷, Human Gut Virome database (HGV), Uncultured Viral Database of Archaeal and Bacteria (GL-UVAB)⁹, GVD¹⁰, GPD and IMG/VR v3¹¹ (detailed in Methods). Based on the archaeal viral detection workflow we developed, we ultimately identified 1,279 nonredundant viral species from the two sources (the longest sequences within each species were selected as the representative).

We have updated the information in lines 118-121.

3) *Ln 184 This is too important to give only a reference. Please specify the criteria for completeness*

Response: Thanks for this suggestion. As a most recently developed software tool, CheckV is an automated pipeline for identifying closed viral genomes and estimating the completeness of genome fragments. CheckV estimates viral completeness by comparing sequences with a large database of complete viral genomes available in the public database¹³. This tool has been widely used in determining the completeness of viruses detected in metagenomic sequencing data. However, we applied this tool to the complete archaeal viral genomes (n=216) downloaded from NCBI's Nucleotide database (GenBank), and around 50% (n=107) were identified as low-quality or undetermined by CheckV¹³. This shows the limitation of this tool in identifying novel archaeal viruses.

The specific criteria for completeness is now added in the Material and Methods section: "Development of archaeal viral detection workflow": "*this most recently developed tool classifies each sequence into one of five quality tiers: complete, high quality (> 90% completeness), medium quality (50-90% completeness), low quality (0-50% completeness) or undetermined quality (no completeness estimate available)*". We have updated this information in lines 493-496.

4) *Ln188 442 were classified as viral, what about the remaining 800?*

Response: We thank for this comment. These comments reveal multiple aspects that we should further clarify.

Here, the 1,279 archaeal viral sequences were identified by the archaeal viral detection pipeline that combines CRISPR matches and archaeal viral signature. We evaluated our data using the various bioinformatic tools. However, these tools were developed based on reference viral genomes available in public datasets, while few human gut archaeal viruses have been available in public database, implying that they are limited in identifying novel archaeal viral sequences. **This was evident by I)** In this revised manuscript, we applied more tools including VirSorter2¹⁶ and DeepVirFinder¹⁷ (score ≥ 0.9 and $p < 0.05$) to the sequences in HGAVD, and the number of the HGAVD sequences (Supplementary Table 6) that were classified as viral sequences increased to 537 in total (line 125). **II)** We compared the HGAVD viral sequences with the proviruses derived from the gut archaeal genomes in UHGG², resulting in 56.5% of the HGAVD viruses sharing $>95\%$ identity with the proviruses. This suggests that many HGAVD sequences that were not taxonomically assigned into known viral order are archaeal viruses. **III)** We also applied these tools to the complete archaeal viral genomes (n=216) downloaded from NCBI Nucleotide database (GenBank), as described previously¹⁸, resulting in 18 viral genomes that were not identified by these tools and around 50% (n=107) were identified as low-quality or undetermined by CheckV¹³. **This showed the limitation of these tools in identifying novel archaeal viruses.**

Thus, for the HGAVD viruses that were not assigned to viruses by these software tools^{16, 17, 31, 32}, the reasons for this can be attributed to **I)** the majority of the archaeal viruses are novel and have no similarity to the references of archaeal viruses deposited in public databases; **II)** due to the fragmental nature of the metagenomic assembled sequences, these identified sequences are only fragments of some whole viral genome sequences that were not identified by these tools. These novel viruses could not be identified by these tools due to the lack of knowledge and must be determined by establishing a culture-dependent method, deep sequencing or long-read sequencing method.

In this study, although these tools were not able to identify the majority of the HGAVD viruses, we conducted the analysis of the archaeal viral sequences related to virus origin, genome quality, functional annotation, taxonomic classification, biogeographic distribution, and host prediction based on the Minimum Information about an Uncultivated Virus Genome (MIUViG) standards¹². Leveraging this comprehensive archaeal viral sequence collection, we provided some glimpses into the human archaeal virome, leading to a better understanding of the human gut ecosystem. **This demonstrates the advantage of the method using CRISPRs-spacers and viral signatures in identifying novel viruses in the human gut.**

5) *Ln 195 Also that the number of viral contigs being present also as proviruses is extremely high (more than 50%) which seems to indicate a very high rate of lysogeny in the gut microbiome*

Response: Many studies of human gut virome imply that the temperate phages are dominant in the human gut virome²⁹. Our study also indicates that the temperate archaeal viruses are dominant in the gut archaeal virome. This was evident by the observations in this study that 56.5% of HGAVD viruses share high identity (>95%) with the proviruses derived from the archaeal genomes and the analysis of 33 HGAVD tailed viruses with a complete genome predicted by CheckV discovered 29 encoding integrase gene or the MazE gene. These genes are predicted important role in regulating phage lytic-lysogen cycle.

6) *Ln 201 This sentence would require clarification and some of the 1279 do not fit into any of the outputs.*

Response: Thanks for the suggestion. We have modified the sentence as following: "With the sequences from the archaeal viral genomes in the database RefSeq and the 1,279 HGAVD viral species, this analysis clustered 735 into 61VC, 391 into outliers (where contigs were assigned to a VC but shared fewer similar proteins than the bulk of the cluster), and 153 into singletons (sequences that did not cluster with any other sequences)" (735+391+153=1,279). (lines 134-138).

7) *Ln 212 Fig 2. Identification of archaeal viruses from the human gut in this work*

Response: Thanks for this suggestion. The figure caption of Fig.2 has been revised as suggested and moved to Fig. 1.

8) *Ln 241 Not sure what is "left panel" why not use the letters?*

Response: We thank the reviewer for catching this error. The "left panel" has been revised as "Fig. 1dI" as suggested.

Response Fig. 11 (Fig. 1d) Protein clustering network of the HGAVD viruses.

9) *Ln 265 The Tanzanian datasets were from some specific cities, ethnicity, lifestyle?*

Response: The Tanzanian datasets were sampled from Hadza people in Africa which are a group of people far away from modern civilization²². The Hadza, a community residing near Lake Eyasi in the central Rift Valley of Tanzania, are among the last remaining populations in Africa. The Hadza people maintain a hunter-gatherer subsistence strategy, relying solely on the wild foods and natural water sources that they find in their region.

10) *Ln303 please explain the difference between M. smithii and M. smithii_A.*

Response: In all this study, all the archaeal genomes were assigned names using the GTDB classification pipeline³. Pasolli et al⁵ recovered 675 genomes of human gut Archaea and reconstructed its phylogeny. They found that more than half of these genomes (n = 487) belonged to the *Methanobrevibacter smithii* species-level genome bin (ID 714), and related but diverged species-level genome bin including 94 genomes was identified (ID 713, 5.6% nucleotide divergence from the *M. smithii* isolate genome). Chibani et al⁶ report the analysis of public archaeal genomes from the human gut, and propose splitting the most abundant archaeal species, *Methanobrevibacter smithii*, into two species-level clades (tentatively named ‘*smithii*’ and ‘*smithii_A*’ according to the GTDB classification³, based on an ANI of only 93.95% between the two representative genomes of *M. smithii* and *M. smithii_A*, as well as information derived from the protein catalogue. *M. smithii_A* and *M. smithii* cannot be distinguished on 16S ribosomal RNA gene sequences, which is most probably the reason for missing this clade separation previously. However, analysis of the *mcrA* gene revealed a consistent difference between these two clades, with an average of 2.15% difference in amino acid sequence.

According to the GTDB classification taxonomy, both *M. smithii* and *M.smithii_A* belong to the genus *Methanobrevibacter_A*. Therefore, we used *Methanobrevibacter_A smithii* and *Methanobrevibacter_A smithii_A* for these two species in this revised manuscript.

11) *Ln 306. Nothing surprising here. Broad host range viruses have been known for long in all domains of life.*

Response: We agree that broad host range viruses have been known for long in all domains of life. We have rephrased the sentence in the revised manuscript.

12) *Ln 337 40% viral genes without matches are to be expected, whether or not they have a function is another issue altogether, the percentage of those should be much higher actually.*

Response: We agree with this reviewer for this comment. In this study, we used a relatively relax cutoff ($<1e-05$, score >50) to assign the gene function. Indeed, when we used a relatively rigorous cutoff value ($<1e-30$, score >50) for this assignment, around 64% viral genes were not assigned functions.

13) *Ln 345 explain “annotated for lysis”.*

Response: We apologize for the confusion. “The proteins annotated for lysis” refers to the proteins annotated with typical viral functional category "lysis-related functions" using consistent method with the study by Nayfach et al.²¹. For example, in this study, N-acetylmuramoyl-L-alanine amidase is an enzyme that catalyzes a chemical reaction that cleaves the link between N-acetylmuramoyl residues and L-amino acid residues in certain cell-wall glycopeptides. The systematic name of this enzyme class is peptidoglycan amidohydrolase (PF01520). And anti-repressor proteins (PF03374) governs the switch from a lysogenic to lytic cycle (The annotated nucleotide sequences (GFF) are available in figshare with the DOI link <https://doi.org/10.6084/m9.figshare.21152404.v3>).

14) *Ln 362 how were they predicted as provirus?*

Response: We predicted the proviral sequences via CheckV¹³, which is an automated pipeline for identifying closed viral genomes, estimating the completeness of genome fragments and removing flanking host DNA from integrated proviruses. CheckV assigns the viral sequences as proviruses if it detects the viral sequences flanked by the genes derived from archaeal or bacterial hosts.

We thank the reviewer again for providing these very insightful comments again.

References

1. Rinke, C. et al. A standardized archaeal taxonomy for the Genome Taxonomy Database. *Nat Microbiol* **6**, 946-959 (2021).
2. Almeida, A. et al. A unified catalog of 204,938 reference genomes from the human gut microbiome. *Nat Biotechnol* **39**, 105-114 (2021).
3. Parks, D.H. et al. A complete domain-to-species taxonomy for Bacteria and Archaea. *Nat Biotechnol* **38**, 1079-1086 (2020).
4. Ciufu, S. et al. Using average nucleotide identity to improve taxonomic assignments in prokaryotic genomes at the NCBI. *Int J Syst Evol Microbiol* **68**, 2386-2392 (2018).
5. Pasolli, E. et al. Extensive Unexplored Human Microbiome Diversity Revealed by Over 150,000 Genomes from Metagenomes Spanning Age, Geography, and Lifestyle. *Cell* **176**, 649-662 e620 (2019).
6. Chibani, C.M. et al. A catalogue of 1,167 genomes from the human gut archaeome. *Nat Microbiol* **7**, 48-61 (2022).
7. Paez-Espino, D. et al. Uncovering Earth's virome. *Nature* **536**, 425-430 (2016).
8. Shkoporov, A.N. et al. The Human Gut Virome Is Highly Diverse, Stable, and Individual Specific. *Cell Host Microbe* **26**, 527-541 e525 (2019).
9. Coutinho, F.H., Edwards, R.A. & Rodriguez-Valera, F. Charting the diversity of uncultured viruses of Archaea and Bacteria. *BMC Biol* **17**, 109 (2019).
10. Gregory, A.C. et al. The Gut Virome Database Reveals Age-Dependent Patterns of Virome Diversity in the Human Gut. *Cell Host Microbe* **28**, 724-740 e728 (2020).
11. Roux, S. et al. IMG/VR v3: an integrated ecological and evolutionary framework for interrogating genomes of uncultivated viruses. *Nucleic Acids Res* **49**, D764-D775 (2021).
12. Roux, S. et al. Minimum Information about an Uncultivated Virus Genome (MIUViG). *Nat Biotechnol* **37**, 29-37 (2019).
13. Nayfach, S. et al. CheckV assesses the quality and completeness of metagenome-assembled viral genomes. *Nat Biotechnol* **39**, 578-585 (2021).
14. Chen, I.A. et al. IMG/M v.5.0: an integrated data management and comparative analysis system for microbial genomes and microbiomes. *Nucleic Acids Res* **47**, D666-D677 (2019).
15. Gregory, A.C. et al. Marine DNA Viral Macro- and Microdiversity from Pole to Pole. *Cell* **177**, 1109-1123 e1114 (2019).
16. Guo, J. et al. VirSorter2: a multi-classifier, expert-guided approach to detect diverse DNA and RNA viruses. *Microbiome* **9**, 37 (2021).
17. Ren, J. et al. Identifying viruses from metagenomic data using deep learning. *Quant Biol* **8**, 64-77 (2020).
18. Graziotin, A.L., Koonin, E.V. & Kristensen, D.M. Prokaryotic Virus Orthologous Groups (pVOGs): a resource for comparative genomics and protein family annotation. *Nucleic Acids Res* **45**, D491-D498 (2017).
19. Huerta-Cepas, J. et al. eggNOG 4.5: a hierarchical orthology framework with improved functional annotations for eukaryotic, prokaryotic and viral sequences. *Nucleic Acids Res* **44**, D286-293 (2016).
20. Pradier, L., Tissot, T., Fiston-Lavier, A.S. & Bedhomme, S. PlasForest: a homology-based random forest classifier for plasmid detection in genomic datasets. *BMC Bioinformatics* **22**, 349 (2021).
21. Nayfach, S. et al. Metagenomic compendium of 189,680 DNA viruses from the human gut microbiome. *Nat Microbiol* **6**, 960-970 (2021).
22. Smits, S.A., Leach, J., Sonnenburg, E. D., Gonzalez, C. G., Lichtman, J. S., Reid, G., ... & Sonnenburg, J. L. Seasonal cycling in the gut microbiome of the Hadza hunter-gatherers of Tanzania. *Science* **357**, 802-806 (2017).
23. Li, R. et al. SOAP2: an improved ultrafast tool for short read alignment. *Bioinformatics* **25**, 1966-1967 (2009).
24. Moissl-Eichinger, C. et al. Archaea Are Interactive Components of Complex Microbiomes. *Trends Microbiol* **26**, 70-85 (2018).
25. Reyes, A., Semenkovich, N.P., Whiteson, K., Rohwer, F. & Gordon, J.I. Going viral: next-generation sequencing applied to phage populations in the human gut. *Nat Rev Microbiol* **10**, 607-617 (2012).

26. Terzian, P. et al. PHROG: families of prokaryotic virus proteins clustered using remote homology. *NAR Genom Bioinform* **3**, Iqab067 (2021).
27. Luo, Y., Pfister, P., Leisinger, T., & Wasserfallen, A. Pseudomurein endoisopeptidases PeiW and PeiP, two moderately related members of a novel family of proteases produced in Methanothermobacter strains. *FEMS Microbiology Letters* **208**, 47-51 (2002).
28. Luo, Y., Pfister, P., Leisinger, T. & Wasserfallen, A. The genome of archaeal prophage PsiM100 encodes the lytic enzyme responsible for autolysis of Methanothermobacter wolfeii. *J Bacteriol* **183**, 5788-5792 (2001).
29. Canchaya, C., Fournous, G. & Brussow, H. The impact of prophages on bacterial chromosomes. *Mol Microbiol* **53**, 9-18 (2004).
30. Simmonds, P. et al. Consensus statement: Virus taxonomy in the age of metagenomics. *Nat Rev Microbiol* **15**, 161-168 (2017).
31. Roux, S., Enault, F., Hurwitz, B.L. & Sullivan, M.B. VirSorter: mining viral signal from microbial genomic data. *PeerJ* **3**, e985 (2015).
32. Ren, J., Ahlgren, N.A., Lu, Y.Y., Fuhrman, J.A. & Sun, F. VirFinder: a novel k-mer based tool for identifying viral sequences from assembled metagenomic data. *Microbiome* **5**, 69 (2017).

REVIEWERS' COMMENTS

Reviewer #1 (Remarks to the Author):

I believe that the authors have significantly improved the manuscript and took on board many of my suggestions. I, however, still have some concerns about the use of some official and unofficial taxonomic names. Here's some further comments to the rebuttal letter:

Comments 1 and 7: "Methanobrevibacter, which includes the type species *Methanobrevibacter ruminantium*, and other four genera with *Methanobrevibacter_D*)."... "As such, in the case of *Methanobrevibacter_A*, based on GTDB, the genus *Methanobrevibacter* has been divided into five genus-level groups: *Methanobrevibacter*, which includes the type species *Methanobrevibacter ruminantium*, and other four genera with alphabetical suffixes (*Methanobrevibacter_A* to *Methanobrevibacter_D*). "*Methanomassiliicoccus_A*" and "*Methanocorpusculum MX-02*" stand for new uncultured archaeal genera in GTDB, respectively."

- I understand the reasoning for using this taxonomic system, however, only officially recognised ICNP names (check <https://lpsn.dsmz.de/> for reference) should be italicised in print. "*Methanobrevibacter*" OR "*Methanobrevibacter ruminantium*" OR "*M. smithii*" should be italicised, while "*Methanobrevibacter_A*", "*Methanomassiliicoccus_A*", "*M. smithii_A*" etc should be written in normal font.

Comment 17: The authors need to make sure that all criteria for selection and filtering of viral sequences listed here are clearly described in the main manuscript.

Reviewer #2 (Remarks to the Author):

The authors have made a thorough effort to address the comments of the three reviewers. The manuscript is significantly improved. However, much to my regret, my principal concern, echoed by reviewer #3, namely, the lack of biological insight that could have been gained through careful analysis of the genes of the discovered archaeal viruses, has not been adequately addressed. Nor

have the authors done enough to move technical details into Methods as was also suggested by two reviewers. As a results, the manuscript remains an unnecessarily difficult read.

Reviewer #3 (Remarks to the Author):

I am satisfied with the authors' responses.

Below, we provide point-by-point responses to reviewers' comments (*italicized, blue*). The corresponding revisions in the manuscript are highlighted in green.

REVIEWERS' COMMENTS

Reviewer #1 (Remarks to the Author):

I believe that the authors have significantly improved the manuscript and took on board many of my suggestions. I, however, still have some concerns about the use of some official and unofficial taxonomic names. Here's some further comments to the rebuttal letter:

Comments 1 and 7: "Methanobrevibacter, which includes the type species Methanobrevibacter ruminantium, and other four genera with Methanobrevibacter_D)."... "As such, in the case of Methanobrevibacter_A, based on GTDB, the genus Methanobrevibacter has been divided into five genus-level groups: Methanobrevibacter, which includes the type species Methanobrevibacter ruminantium, and other four genera with alphabetical suffixes (Methanobrevibacter_A to Methanobrevibacter_D). "Methanomassiliicoccus_A" and "Methanocorpusculum MX-02" stand for new uncultured archaeal genera in GTDB, respectively."

- I understand the reasoning for using this taxonomic system, however, only officially recognised ICNP names (check <https://lpsn.dsmz.de/> for reference) should be italicised in print. "Methanobrevibacter" OR "Methanobrevibacter ruminantium" OR "M. smithii" should be italicised, while "Methanobrevibacter_A", "Methanomassiliicoccus_A", "M. smithii_A" etc should be written in normal font.

Response: Thanks for the suggestion. We have revised "not officially recognised ICNP names" to normal font throughout the paper.

Comment 17: The authors need to make sure that all criteria for selection and filtering of viral sequences listed here are clearly described in the main manuscript.

Response: All criteria for selection and filtering of viral sequences listed in response to Comment 17 are included in the main manuscript as suggested.

We thank this reviewer again for providing these very helpful comments again.

Reviewer #2 (Remarks to the Author):

The authors have made a thorough effort to address the comments of the three reviewers. The manuscript is significantly improved. However, much to my regret, my principal concern, echoed by reviewer #3, namely, the lack of biological insight that could have been gained through careful analysis of the genes of the discovered archaeal viruses, has not been adequately addressed. Nor have the authors done enough to move technical details into Methods as was also

suggested by two reviewers. As a result, the manuscript remains an unnecessarily difficult read.

Response: We thank this reviewer for recognizing our efforts in improving the manuscript.

We have made a substantial analysis of the genes of the discovered archaeal viruses in the revised manuscript and the biological information has been updated in the section of Results “*Archaeal virus genomes encode an extensive functional repertoire*”. We feel sorry that this was not addressed in the last response letter. Here, we further clarify for this concern.

These identified archaeal viruses enable us to explore the functional potential of the archaeal virome in the human gut. To do this, we identified 97,208 protein-coding genes on the representative sequences of these 1,279 viral species. Overall, 40% (n=39,268) of the viral genes did not have significant matches (cutoff: e-value < 1e-5, score > 50) in the Pfam(v32) database¹ and were not assigned to any biological functions. When we used a relatively rigorous cutoff value (<1e-30, score>50) for annotation in the Pfam database, around 64% HGAVD viruses genes were not assigned functions. Only 10.8% and 17.4% of these genes had hits in pVOG² and PHROG³, respectively. Comparative genomic analysis on the representative sequences selected for each VC of the complete HGAVD sequences (Fig. 4c) indicated that they were shown divergent in genomic sequence and most of the genes encoding for hypothetical proteins (please see in lines 283-329, Fig.4 and Supplementary Fig. 7).

Nevertheless, based on the analysis of the genes encoded by the identified archaeal viral sequences, we found that **1)** numerous functional genes were encoded by these archaeal viruses (Fig.4a and Supplementary Fig. 7); in particular, **2)** the viruses of *Methanobrevibacter_A smithii* contained the most functional diversity with proteins homologous to 1,034 different kinds of tailed-virus-specific proteins in the Pfam database (Fig.4b); **3)** a diversity of the tailed archaeal viruses and new archaeal viral taxonomies were shown based on the phylogenetic analysis of large subunit terminase genes encoded by these archaeal viruses (Fig. 3d); **4)** a gene encoding the protein homologous to pseudomurein endoisopeptidase (PeiW) frequently occurred on many viral genomes (n=150) (Supplementary Fig.8). The prototype PeiW is found in the archaeal prophage psiM100 as an autolytic enzyme produced by the thermophilic methanoarchaeon *Methanothermobacter wolfeii* to cleave pseudomurein cell-wall sacculi of archaeal methanogens⁴. The phylogenetic analysis of PeiW revealed that except for the viruses of *M. wolfeii*, other archaeal viruses also were the carrier of *peiW*, such as the viruses of *Methanobrevibacter_A smithii* and *Methanobrevibacter olleyae* (Supplementary Fig. 8); **5)** in the analysis of these complete *Caudoviricetes* viral genomes, 29 of 33 encoded the genes for phage integrase protein and 10 genomes encoded proteins (Supplementary Fig. 11-13) belonging to the antitoxin MazE superfamily (Supplementary Fig.10) that plays an important role in regulating the viral lysogenic-lytic cycle; **6)** other genes with functions were shown in Fig.4a.

We conducted a substantial revision in moving the technical details from Results and Discussion to Methods. In summary, 1) The subsection of Results “*Identification of archaeal genomic contigs from the metagenomes expands the archaeal diversity in the human gut*” and the related information have been moved to the Supplementary information; 2) Many technical details about the identification of the archaeal viruses and taxonomic assignment have been moved to Methods and updated; 3) The specific criteria for identifying archaeal viruses and the version information of

the software tools have been added in the Methods section; 4) We show some necessary technical information in the Results section to make the readers' reading smooth.

We thank this reviewer again for carefully reading our manuscript and providing helpful comments.

Reviewer #3 (Remarks to the Author):

I am satisfied with the authors' responses.

Response: We thank this reviewer for the constructive comments.

1. El-Gebali, S. et al. The Pfam protein families database in 2019. *Nucleic Acids Res* **47**, D427-D432 (2019).
2. Graziotin, A.L., Koonin, E.V. & Kristensen, D.M. Prokaryotic Virus Orthologous Groups (pVOGs): a resource for comparative genomics and protein family annotation. *Nucleic Acids Res* **45**, D491-D498 (2017).
3. Terzian, P. et al. PHROG: families of prokaryotic virus proteins clustered using remote homology. *NAR Genom Bioinform* **3**, lqab067 (2021).
4. Luo, Y., Pfister, P., Leisinger, T., & Wasserfallen, A. Pseudomurein endoisopeptidases PeiW and PeiP, two moderately related members of a novel family of proteases produced in *Methanothermobacter* strains. *FEMS Microbiology Letters* **208**, 47-51 (2002).